# EVEREST:
# AN EVIDENTIAL, TAIL-AWARE TRANSFORMER FOR RARE-EVENT TIME-SERIES FORECASTING

**Antanas Žilinskas**
Imperial College London
antanas.zilinskas.research@gmail.com

**Robert N. Shorten**
Imperial College London
r.shorten@imperial.ac.uk

**Jakub Mareček**
Czech Technical University in Prague, Faculty of Electrical Engineering
jakub.marecek@fel.cvut.cz

## ABSTRACT

Forecasting rare events in multivariate time-series data is challenging due to severe class imbalance, long-range dependencies, and distributional uncertainty. We introduce EVEREST, a transformer-based architecture for probabilistic rare-event forecasting that delivers calibrated predictions and tail-aware risk estimation, with auxiliary interpretability via attention-based signal attribution. EVEREST integrates four components: (i) a learnable attention bottleneck for soft aggregation of temporal dynamics; (ii) an evidential head for estimating aleatoric and epistemic uncertainty via a Normal–Inverse–Gamma distribution; (iii) an extreme-value head that models tail risk using a Generalized Pareto Distribution; and (iv) a lightweight precursor head for early-event detection. These modules are jointly optimized with a composite loss (focal loss, evidential NLL, and a tail-sensitive EVT penalty) and act only at training time; deployment uses a single classification head with no inference overhead (approximately 0.81M parameters). On a decade of space-weather data, EVEREST achieves state-of-the-art True Skill Statistic (TSS) of 0.973/0.970/0.966 at 24/48/72-hour horizons for C-class flares. The model is compact, efficient to train on commodity hardware, and applicable to high-stakes domains such as industrial monitoring, weather, and satellite diagnostics. Limitations include reliance on fixed-length inputs and exclusion of image-based modalities, motivating future extensions to streaming and multimodal forecasting.

## 1 INTRODUCTION

Rare, high-impact events in multivariate time series pose a central challenge in machine learning, with direct implications for space weather, industrial monitoring, power systems, and satellite health. Models must contend with three factors simultaneously: severe class imbalance, long-range temporal dependencies that dilute early precursors, and the need for calibrated probabilities and explicit tail-risk assessment in thresholded, operational decision-making. Standard objectives under-weight extremes, and average losses (e.g., cross-entropy) provide little guidance in settings where false negatives are disproportionately costly.

These challenges are threefold. First, extreme rarity and long horizons make discriminative learning difficult: positive sequences are sparse, contexts are long, and recent long-horizon architectures—from frequency-based decompositions (Zhou et al., 2022) to patching (Nie et al., 2023) and modern convolutions (Luo & Wang, 2024)—improve aggregation but do not directly address calibration under rarity. Prior forecasting models such as recurrence-based approaches (Liu et al., 2019), hybrid CNN–Transformer designs (Sun et al., 2022), and flare-specific architectures (Abduallah et al., 2023) achieve strong discrimination but provide limited tools for calibrated uncertainty or tail behaviour. Second, high-stakes applications require reliable probabilities: miscalibration degrades

operational utility, especially for rare-event thresholds where uncertainty decomposition (aleatoric vs. epistemic) matters for decision support (Sensoy et al., 2018; Amini et al., 2020; van Amersfoort et al., 2020). Third, catastrophic outcomes occupy the far tail, where standard losses provide little gradient signal; modelling exceedances beyond a high quantile is well studied in extreme-value theory (EVT) (Coles, 2001; de Haan & Ferreira, 2006), but rarely coupled with neural sequence models.

Here we show that these three challenges can be jointly addressed in a single, compact transformer. We introduce EVEREST, which integrates a learnable single-query attention bottleneck for long-range temporal aggregation with three training-only auxiliaries: an evidential Normal–Inverse–Gamma (NIG) head that regularises logit calibration, an EVT exceedance head that shapes far-tail behaviour using a Generalised Pareto penalty, and a lightweight precursor head that imposes anticipatory supervision. These components act solely during training; inference uses a single classification head, so runtime cost is identical to a standard transformer of comparable size. Across nine SHARP–GOES solar-flare tasks (2010–2023), EVEREST achieves state-of-the-art TSS (e.g., $0.973/0.970/0.966$ for $\geq$C at 24/48/72 h and $0.907/0.936/0.966$ for $\geq$M5) with strong calibration (e.g., M5–72 h ECE $= 0.016$). The model also transfers without architectural changes to an industrial anomaly dataset (SKAB), reaching F1 $= 98.16\%$ and TSS $= 0.964$.

The design is intentionally practical and general: a single encoder and bottleneck aggregate temporal evidence; training-only auxiliaries regularise the shared representation; inference remains lightweight (0.81M parameters, no auxiliary heads). We provide extensive ablations quantifying the marginal utility of each component and analyse reliability, tail sensitivity, and thresholded decision performance. We further provide systematic sensitivity analyses over evidential and EVT loss weights and exceedance quantiles, showing that performance is robust across wide hyperparameter ranges—consistent with these terms acting as regularisers rather than fragile knobs.

The remainder of the paper situates the method among recent time-series transformers, calibration approaches, and EVT-based models; details the backbone, bottleneck, auxiliaries, and composite loss; and presents results, ablations, robustness studies, and limitations. We conclude by discussing broader applicability in scientific and industrial forecasting, where compact, calibrated, tail-aware models are increasingly required.

## 2 RELATED WORK

Our work draws on a long history of work scattered across multiple research communities. Let us present these in turn:

**Rare-event time series and imbalance.** Forecasting rare events in multivariate time series requires handling both severe class imbalance and long temporal dependencies. Early approaches often treated each window independently, combining hand-crafted features with a classifier, whereas modern deep models such as TCNs and Transformers exploit sequential structure more effectively. Cost-sensitive objectives like focal loss address imbalance without altering the data distribution, while aggressive oversampling can introduce temporal artefacts or leakage, motivating loss-based rather than data-based rebalancing strategies for scientific and operational settings. Recent supervised pipelines report strong solar-flare discrimination at 24–72 h horizons, including CNN/RNN hybrids and flare-specific Transformers (Liu et al., 2019; Sun et al., 2022; Abduallah et al., 2023). We report results on the same SHARP–GOES benchmark.

**Transformers for time series.** Transformers have become highly competitive in time-series forecasting, but naïve self-attention incurs $\mathcal{O}(T^2)$ cost. Recent architectures address this by restructuring temporal information through patch/token re-organization with channel-first encoders (Nie et al., 2023), frequency- or decomposition-based long-horizon modules (Zhou et al., 2022), or inverted designs that summarize time before mixing channels (Liu et al., 2024). Pure-convolutional models can rival attention on long sequences at lower computational cost (Luo & Wang, 2024). Our single-query attention bottleneck provides a lightweight, task-conditioned global aggregator in this design space, conceptually closer to attention pooling and global token mechanisms (Ilse et al., 2018; Lee et al., 2019) than to full self-attention over all time steps.

**Calibration and evidential learning.** High-stakes forecasting requires not only discriminative accuracy but also reliable probabilities. Beyond common metrics such as TSS or AUPRC, calibration metrics (ECE, Brier score) directly inform operational thresholding. Post-hoc strategies such as temperature scaling (Guo et al., 2017) can improve marginal reliability but cannot recover input-conditional epistemic uncertainty. Deterministic OOD surrogates and deep ensembles (van Amersfoort et al., 2020; Lakshminarayanan et al., 2017) offer stronger uncertainty estimates at higher computational cost. Evidential methods instead learn closed-form distributional parameters, e.g. Dirichlet for classification or Normal–Inverse–Gamma (NIG) for regression, enabling uncertainty decomposition without Monte Carlo sampling (Sensoy et al., 2018; Amini et al., 2020). Complementary developments in conformal prediction provide distribution-shift–robust error control in time series (Ding et al., 2023). We adopt an evidential NIG head directly on the logit to regularise calibration during training.

**Tail risk and EVT in machine learning.** Standard objectives under-weight catastrophic extremes because they allocate little gradient mass to high-quantile regions. The peaks-over-threshold framework from extreme value theory (EVT) offers a principled treatment of distribution tails using Generalized Pareto exceedances (Coles, 2001; de Haan & Ferreira, 2006). Recent work has applied EVT to extreme-event prediction directly on time-series signals (Kozerawski et al., 2022). Our approach differs in that the GPD is fitted to *logit exceedances*, allowing EVT to act as a training-time tail-shaping regulariser rather than a post-hoc or residual-based tail estimator.

**Auxiliary/precursor supervision and multi-task learning.** Auxiliary tasks can improve a primary task by regularising a shared backbone, even when auxiliary heads are removed at inference, as demonstrated in classical and modern multi-task learning (Caruana, 1997; Standley et al., 2020). Contrastive forecasting objectives (Makansi et al., 2021) similarly inject early-event structure by contrasting imminent-event windows against quiescent ones. Although EVEREST does not employ contrastive learning, its lightweight precursor head plays a related role by encouraging anticipatory representations; integrating a contrastive variant is a natural direction for future work.

**Industrial anomaly benchmarks.** The SKAB dataset provides multivariate valve traces commonly used in time-series anomaly detection (Filonov et al., 2020). Among strong baselines, TranAD reports leading performance across datasets (Tuli et al., 2022). For comparability, we adopt the same windowing, labelling, and evaluation protocol used in published work.

**Gap and positioning.** Most prior work improves sequence encoders *or* enhances calibration, but rarely addresses calibration and tail sensitivity jointly in a compact architecture. EVEREST fills this gap by combining (i) a single-query attention bottleneck for long-context aggregation, (ii) an evidential NIG head for closed-form logit calibration, and (iii) an EVT exceedance penalty to emphasise extremes—all optimised jointly while retaining a single classification head at test time with no inference overhead.

## 3 METHOD

We consider binary rare–event forecasting on multivariate time series. Each example is a window $X \in \mathbb{R}^{T \times F}$, containing $T$ time steps and $F$ features, with label $y \in \{0, 1\}$ indicating whether an event occurs within a fixed forecast horizon. The model outputs a logit $l \in \mathbb{R}$ and a probability $\hat{p} = \sigma(l) \in [0, 1]$, which is compared to a decision threshold $\tau$ to produce an alert. We report skill with the True Skill Statistic (TSS) and assess reliability with the Brier score and Expected Calibration Error (ECE).

For clarity, we summarise the main notation used in this section. Encoder layer $l$ outputs hidden states $H^{(l)} = \{h_t^{(l)}\}_{t=1}^{T}$, which the attention bottleneck pools into a single representation $z$. The evidential head outputs Normal–Inverse–Gamma parameters $(\mu, v, \alpha, \beta)$ over the logit, and the EVT head predicts Generalised Pareto parameters $(\xi, \sigma)$ for exceedances above a high quantile $u$. The composite loss is controlled by non-negative coefficients $(\lambda_f, \lambda_e, \lambda_t, \lambda_p)$. A full notation table is provided in Appendix K.

### 3.1 ARCHITECTURE OVERVIEW

The network comprises four stages: (i) an input embedding with scaled positional encoding, (ii) a $6\times$ Transformer encoder, (iii) a single-query attention bottleneck that aggregates the sequence into a single latent vector $z$, and (iv) a shallow shared MLP (128-d) from which four parallel heads branch: a primary binary classification logit (used at inference) and three training-only auxiliaries—evidential (NIG), EVT (GPD) exceedance, and a lightweight precursor head.

Unless explicitly stated otherwise, deployment uses only the classification head in a single forward pass. The evidential, EVT, and precursor heads act as training-time auxiliaries that regularise the shared representation and can be evaluated offline for diagnostics, but are never required for test-time decisions.

In the embedding and transformer backbone, cf. (i) and (ii) above, raw inputs $X$ are projected to $d$-dimensional tokens and combined with sinusoidal positional codes scaled by a learnable global factor $\alpha$:
$$h_0 = \text{LN}(W_{\text{emb}}X + b_{\text{emb}}), \qquad H^{(0)} = \text{Drop}(h_0 + \alpha \cdot \text{PE}),$$
where $W_{\text{emb}} \in \mathbb{R}^{d \times F}$ and $b_{\text{emb}} \in \mathbb{R}^d$ are learned.

We apply $L=6$ encoder blocks with multi-head self-attention and position-wise feed-forward networks:
$$\tilde{H}^{(l)} = \text{LN}\big(H^{(l-1)} + \text{Drop}[\text{MHA}(H^{(l-1)})]\big), \quad H^{(l)} = \text{LN}\big(\tilde{H}^{(l)} + \text{Drop}[\text{FFN}(\tilde{H}^{(l)})]\big),$$
for $l = 1, \ldots, 6$. The reference setting (§4) uses embedding dimension $d=128$, $L=6$ layers, $H=4$ attention heads, FFN width 256, and dropout $p=0.20$.

In the attention bottleneck, cf. (iii) above, one undertakes temporal focussing. Let $\mathbf{H} = [h_1, \ldots, h_T] \in \mathbb{R}^{d \times T}$ denote the final encoder states and $w \in \mathbb{R}^d$ a learned scorer. We compute a single soft attention distribution over time and the pooled vector

$$\alpha_t = \text{softmax}_t\big(w^\top h_t\big), \qquad z = \sum_{t=1}^{T} \alpha_t\, h_t, \quad w \in \mathbb{R}^d.$$

This *single-query* bottleneck adds only $+d$ parameters and $\mathcal{O}(Td)$ flops, yet concentrates capacity on weak, distributed precursors that global average pooling tends to dilute. In ablations (§5), replacing the bottleneck with mean pooling substantially reduces skill (e.g., $\Delta\text{TSS} = +0.427$ on the hardest M5–72 h task).

Finally, the pooled representation $z$ feeds four parallel linear heads that share the backbone and MLP parameters, cf. (iv) above.

**Classification head.** The primary head produces a scalar logit
$$l = W_{\text{clf}}z + b_{\text{clf}},$$
with probability $\hat{p} = \sigma(l)$ used for all thresholded decisions.

**Evidential (NIG) head.** The evidential head predicts parameters $(\mu, v, \alpha, \beta)$ of a Normal–Inverse–Gamma distribution over the logit $l$, and minimises a closed-form evidential objective, yielding analytic predictive mean and variance without Monte Carlo sampling. This acts as a Bayesian surrogate that regularises logit-level uncertainty. In ablations it primarily improves discrimination on the hardest tasks (e.g., $\Delta\text{TSS} = +0.064$ on M5–72 h; §5) while maintaining low ECE.

**EVT (GPD) head.** The EVT head predicts Generalised Pareto parameters $(\xi, \sigma)$ for logit exceedances above a high batchwise quantile $u$ (90% by default). For logits $\{l_i\}$ in a mini-batch, we form exceedances $\{l_i - u : l_i > u\}$ and maximise the GPD log-likelihood with a small stability regulariser on $(\xi, \sigma)$ to avoid degenerate tails. This shifts gradient mass towards the risky upper tail and improves rare-event sensitivity.

**Precursor (auxiliary) head.** The precursor head reuses the same binary label and is trained via binary cross-entropy as an auxiliary objective providing *anticipatory supervision*. It is not used at inference. In ablations, removing it degrades M5–72 h TSS by $-0.650$ (§5), indicating that early supervision materially shapes the backbone.

## 3.2 COMPOSITE LOSS AND TRAINING SCHEDULE

The training objective unifies four complementary criteria—discrimination, calibration, tail aware-ness, and anticipatory supervision—within a single composite loss:

$$\mathcal{L} = \lambda_f \, \mathcal{L}_{\text{focal}} + \lambda_e \, \mathcal{L}_{\text{evid}} + \lambda_t \, \mathcal{L}_{\text{evt}} + \lambda_p \, \mathcal{L}_{\text{prec}}.$$

Only the *relative* values of $(\lambda_f, \lambda_e, \lambda_t, \lambda_p)$ matter, since any common scaling leaves the optimiser invariant; in practice we parameterise these as normalised weights (up to a shared scale factor) and use $(\lambda_f, \lambda_e, \lambda_t, \lambda_p) = (0.8, 0.1, 0.1, 0.05)$ as the reference setting. The decreasing values are inspired by the Information Bottleneck principle (Tishby et al., 2000): the encoder compresses inputs $X$ into a latent $Z$ while maximising mutual information $I(Z; Y)$ with the event label. Each loss term targets a distinct aspect of this balance: $\mathcal{L}_{\text{focal}}$ improves separation under extreme rarity, $\mathcal{L}_{\text{evid}}$ regularises predictive entropy, $\mathcal{L}_{\text{evt}}$ reallocates capacity toward tail exceedances, shaping the heavy tail, and $\mathcal{L}_{\text{prec}}$ enriches $I(Z; Y)$ with anticipatory structure. Together, they yield an encoder that balances predictive skill with uncertainty fidelity under extreme rarity, where the auxiliary heads provide calibrated and tail-sensitive regularisation:

- Focal discrimination: The focal term $\mathcal{L}_{\text{focal}}$ addresses class imbalance by re-weighting mis-classified examples according to their difficulty. With focusing parameter $\gamma$, it emphasises hard rare-event examples:

$$\mathcal{L}_{\text{focal}} = -\frac{1}{N} \sum_i \left[ (1 - \hat{p}_i)^\gamma y_i \log \hat{p}_i + \hat{p}_i^\gamma (1 - y_i) \log(1 - \hat{p}_i) \right].$$

  We anneal $\gamma : 0 \to 2$ linearly over the first 50 epochs, initially allowing broad exploration and later sharpening emphasis on difficult rare-event instances.

- Evidential calibration: The evidential term $\mathcal{L}_{\text{evid}}$ learns NIG parameters over the logit, in-ducing a predictive distribution with closed-form mean and variance. This encourages the model to represent epistemic and aleatoric uncertainty at the logit level, providing a cali-brated probability surface without sampling. In practice, ablations show small effects on ECE but consistent gains in TSS on the most imbalanced tasks (§5).

- Tail emphasis via EVT: The EVT term $\mathcal{L}_{\text{evt}}$ fits a Generalised Pareto Distribution to logit exceedances above a high quantile $u$. For a batch of logits $\{l_i\}$, exceedances $\{x_i = l_i - u : l_i > u\}$ are modelled via

$$\Pr(L > u + x \mid L > u) \approx \left( 1 + \tfrac{\xi x}{\sigma} \right)^{-1/\xi},$$

  with $(\xi, \sigma)$ predicted by the EVT head. Maximising the GPD log-likelihood reallocates gradient signal to rare, high-risk predictions, aligning optimisation with extreme-value the-ory and improving sensitivity in the far tail.

- Precursor supervision: The precursor term $\mathcal{L}_{\text{prec}}$ applies binary cross-entropy to the precur-sor head using the same label $y$. It acts as anticipatory supervision, encouraging the latent $Z$ to encode early discriminative cues rather than only near-term features. From the IB perspective, it enriches $I(Z; Y)$ by regularising $Z$ toward features predictive of both early and late outcomes.

As discussed in §5, sensitivity analyses over these weights and the EVT quantiles show that our performance is robust over a wide range of hyperparameters, consistent with the auxiliaries acting as regularisers rather than fragile knobs. All four losses act only at training time; deployment uses the classification head $\hat{p} = \sigma(l)$, with uncertainty and tail diagnostics from the evidential and EVT heads evaluated offline if desired.

**Computational footprint.** At the reference configuration, EVEREST has approximately $8.14 \times 10^5$ parameters and $1.66 \times 10^7$ FLOPs per window; the six-layer backbone accounts for more than $97\%$ of both, while the attention bottleneck adds only $+d$ parameters. A full per-module budget and a comparison to *SolarFlareNet* (Abduallah et al., 2023) are provided in Appendix A.

## 4 EXPERIMENTAL SETUP

### 4.1 DATASETS AND SPLITS

**Solar flares (SHARP–GOES).** We adopt the SHARP–GOES protocol and splits consistent with prior work (Abduallah et al., 2023): SHARP vector-magnetogram parameters aligned to GOES flare labels across Solar Cycle 24–25, with standard quality masks (QUALITY=0, $|\text{CMD}| \leq 70°$, observer radial-velocity filter) applied before windowing. We use the same nine SHARP parameters and the same window construction for 24/48/72 h horizons. To prevent leakage, we use the identical HARPNUM-stratified train/validation/test split; the resulting per-horizon, per-class counts are consolidated in Appendix B (Table 5). All preprocessing (normalization, cadence handling, label alignment) follows that setup to ensure 1:1 comparability.

**SKAB (industrial transfer).** We evaluate cross-domain transfer on the Skoltech Anomaly Benchmark (SKAB) (Filonov et al., 2020) using fixed-length windows (stride two), stacked raw+diff channels, chronological 70/15/15 splits, and standardization fitted on train only. We do not apply oversampling or task-specific loss reweighting. TranAD is the strongest published reference (Tuli et al., 2022). Full data-processing protocol, model configuration, and the complete results/comparisons are provided in Appendix C (Tables 6, 7).

### 4.2 METRICS AND EVALUATION PROTOCOL

**Primary and secondary metrics.** Our primary discrimination metric is the *True Skill Statistic* (TSS),

$$\text{TSS} = \frac{\text{TP}}{\text{TP+FN}} - \frac{\text{FP}}{\text{FP+TN}},$$

reported at the task-specific operating threshold $\tau^\star$ (below). We also report Precision/Recall/F1, AUROC and PR-AUC for ranking quality, and the *Brier score* for probabilistic accuracy. Reliability is quantified via *Expected Calibration Error* (ECE) with equal-frequency binning (15 bins).

**Operating thresholds and cost sensitivity.** Decision thresholds are selected by grid search over $\tau \in \{0.10, 0.11, \ldots, 0.90\}$ using the balanced score (40% TSS, 20% F1, 15% Precision, 15% Recall, 10% Specificity). For sensitivity to asymmetric costs, we complement this with a cost–loss sweep (e.g., $C_{\text{FN}}{:}C_{\text{FP}}{=}20{:}1$) and report the minimum-cost threshold in §5 alongside the balanced operating point.

### 4.3 TRAINING DETAILS AND HPO

All models are trained in PyTorch with automatic mixed precision (AMP), AdamW ($\beta_1{=}0.9, \beta_2{=}0.999$), cosine-decayed learning rate, gradient-norm clipping (1.0), and the composite objective from §3 with $\boldsymbol{\lambda}{=}(0.8, 0.1, 0.1, 0.05)$ and focal $\gamma$ annealed $0{\rightarrow}2$ over the first 50 epochs. Hyper-parameter optimization follows the three-stage protocol (Sobol scan $\rightarrow$ Optuna refinement $\rightarrow$ confirmation), limited to the six knobs that explained the bulk of validation-TSS variance: embedding width $d$, encoder depth $L$, dropout $p$, focal $\gamma$, peak LR $\eta_{\max}$, and batch size $B$. The search priors and the final chosen configuration are in Appendix D; per-scenario optima are tabulated in Appendix D.4.

**Statistical protocol.** For each threshold–horizon task we train five seeds and report means with 95% CIs via $10^4$-draw bootstrap on the held-out test set, stratified by NOAA active-region identifier to preclude temporal leakage. Operating thresholds are selected by a grid over $\tau \in \{0.10, \ldots, 0.90\}$ (step 0.01) using a balanced score (40% TSS, 20% F1, 15% Precision, 15% Recall, 10% Specificity); unless stated, headline metrics use the task-specific $\tau^\star$ from this procedure. The composite-loss weights and related hyperparameters are fixed across all tasks. We also filter obviously failed runs and report detectable effects (e.g., $\Delta\text{TSS} \geq 0.02$) alongside $p$-values from the bootstrap test.

### 4.4 FIGURES AND TABLES FOR REPRODUCIBILITY

To keep the setup self-contained within the page budget, we reuse the same artefacts and protocol as our released implementation:

- **SHARP feature list and motivations** (Table 4): the nine input parameters with brief physical rationale.
- **Dataset distribution** (Table 5): counts per horizon, class, and split under `HARPNUM` stratification.
- **CMD filtering diagram** (Fig. 2): effect of the $|\text{CMD}| \leq 70°$ mask on the usable sequence pool during solar data pre-processing.

## 5 RESULTS

### 5.1 HEADLINE PERFORMANCE

We compare against three published forecasters that span the main model families on SHARP–GOES: (1) an LSTM recurrent predictor (Liu et al., 2019), (2) a 3D-CNN (Sun et al., 2022), and (3) SolarFlareNet (Abduallah et al., 2023), the strongest published baseline. Brief architectural summaries and reproducibility details are provided in Appendix L. EVEREST shows large TSS gains across horizons, with especially strong improvements for rare M5 events. All nine tasks exceed the reported baseline TSS values (Table 1). Table 12 reports bootstrapped metrics; EVEREST delivers consistently high discrimination for common C-class events (TSS $\geq 0.966$ at all horizons) and strong performance for rarer M and M5 classes. See §5.2 for calibration diagnostics, and Appendix E for full per-task results and operating thresholds.

### 5.2 CALIBRATION AND RELIABILITY

We report calibration with Brier score and Expected Calibration Error (ECE; 15 equal-frequency bins) alongside TSS. On the most imbalanced task (M5–72 h) we obtain ECE = 0.016 with a near-diagonal reliability curve; similar trends hold for C–72 h and M–72 h. Diagnostics use the same seeds, splits, and binning as the headline metrics in Table 12. Full reliability diagrams are provided in Appendix F.

### 5.3 DECISION ANALYSIS UNDER ASYMMETRIC COSTS

Operational use often values missed-event costs far above false alarms. For M5–72 h, a cost–loss sweep with $C_{\text{FN}}{:}C_{\text{FP}}$=20:1 yields a minimum-cost threshold of $\tau^{\star} = 0.240$, distinct from the balanced-score $\tau = 0.460$. Figure 1 illustrates the trade-off; the corresponding confusion matrices are in Appendix G.

**By threshold class.** $\geq$**C:** TSS remains within 0.973/0.970/0.966 (24/48/72 h), with precision 0.994/0.993/0.992 and minor horizon decay ($\Delta$TSS= 0.007 from 24 h to 72 h). $\geq$**M:** Despite stronger imbalance, TSS reaches 0.898/0.920/0.906 with recall $\geq 0.908$; precision gains with horizon ($0.728 {\rightarrow} 0.834$). $\geq$**M5:** For the rarest events, TSS is 0.907/0.936/0.966 with tight CIs and the best ECE (e.g., 0.016 at 72 h).

**Comparison to prior work.** Table 1 summarizes TSS versus reported baselines. Our reported scores are higher than published baseline values (e.g., +0.251 TSS for $\geq$C–48 h and +0.237 for $\geq$M5–72 h). Significance testing is applied within our models.

This explicit operating-point choice addresses decision relevance under asymmetric costs without retraining, and the full confusion matrices are provided in Appendix G.

### 5.4 ABLATIONS

A leave-one-component-out suite (five seeds each) quantifies the marginal utility of each module on the hardest task (M5–72 h). Headline effect sizes are:

- **Attention bottleneck:** +0.427 TSS over mean pooling.
- **EVT head:** +0.285 TSS with major extreme-Brier gains.
- **Evidential NIG head:** +0.064 TSS with lower ECE.

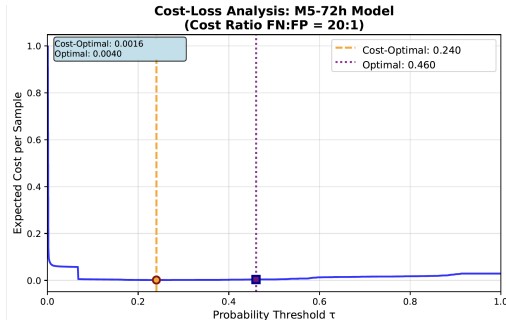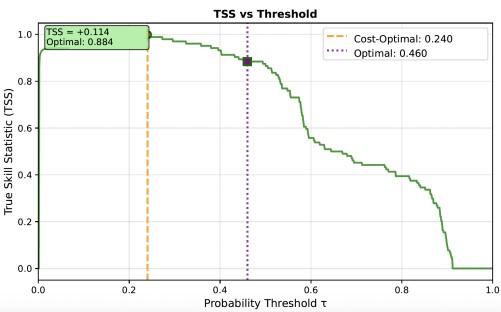

Figure 1: Cost–loss analysis for the M5–72 h model under asymmetric costs ($C_{\mathrm{FN}}{:}C_{\mathrm{FP}} = 20{:}1$). The left panel shows the cost curve; the right panel highlights the minimum-cost threshold $\tau^{\star} = 0.240$ versus the balanced-score threshold $\tau = 0.460$.

Table 1: TSS performance across flare thresholds and horizons. Bold indicates the best performance within each horizon. Reported values for EVEREST are mean (standard deviation) over 5 seeds.

| Method | Horizon | $\geq$C | $\geq$M | $\geq$M5.0 |
|---|---|---|---|---|
| Liu et al. (2019) | 24h | 0.612 | 0.792 | 0.881 |
| Sun et al. (2022) | 24h | 0.756 | 0.826 | – |
| Abduallah et al. (2023) | 24h | 0.835 | 0.839 | 0.818 |
| | 48h | 0.719 | 0.728 | 0.736 |
| | 72h | 0.702 | 0.714 | 0.729 |
| **EVEREST** | 24h | **0.973** (0.001) | **0.898** (0.011) | **0.907** (0.025) |
| | 48h | **0.970** (0.001) | **0.920** (0.007) | **0.936** (0.021) |
| | 72h | **0.966** (0.001) | **0.906** (0.012) | **0.966** (0.024) |

- **Composite schedule:** $+0.045$ TSS from $\gamma$ annealing and stable joint training.

Removing the precursor auxiliary degrades performance by $-0.650$ TSS, showing that anticipatory supervision materially shapes the backbone even though it is discarded at inference. Mixed-precision (AMP) was also indispensable: FP32 runs diverged or underperformed. In addition, a $5{\times}5$ log-scale sweep over the evidential and EVT loss weights shows that both TSS and ECE remain stable across wide regions of the $(\lambda_{\mathrm{evid}}, \lambda_{\mathrm{evt}})$ grid, and an EVT quantile sweep over $u \in \{0.85, 0.90, 0.95\}$ on the hardest task (M5–72 h) yields TSS in $[0.903, 0.932]$ and ECE in $[0.0119, 0.0164]$ across all runs, indicating insensitivity to the precise exceedance threshold. Full per-variant metrics, significance tests, calibration effects, the $\lambda$-sensitivity heatmaps, and EVT-quantile analysis are consolidated in Appendix H.

## 5.5 Interpretability

Saliency analysis highlights how EVEREST differentiates between prediction outcomes. True positives show coordinated increases in USFLUX and MEANGAM in the final hours before the forecast horizon, consistent with flux emergence and field-inclination steepening. True negatives and false positives exhibit flatter or noisier signatures. Confidence-stratified TP cases show that gradients are strongest when predictive confidence is high. Full gradient visualisations are provided in Appendix I.

## 5.6 Prospective Case Study

We evaluate EVEREST on the unseen 6 Sep 2017 X9.3 flare (NOAA AR 12673), the largest event of Solar Cycle 24. Data from 3–7 September 2017 were excluded from training and threshold calibration. The probability trace and lead-time statistics are provided in Appendix J (Figure 11 and Table 18).

## 5.7 CROSS-DOMAIN TRANSFER: SKAB

With the architecture unchanged, EVEREST achieves mean TSS = 0.964 and F1 = 98.16% on SKAB (Filonov et al., 2020). We include SKAB because it is multivariate, rare-event–oriented, and widely used in anomaly detection; baseline results (e.g., TranAD)(Tuli et al., 2022). Full valve-level metrics and calibration diagnostics are in Appendix C.

## 5.8 EFFICIENCY SNAPSHOT

Training uses AMP and the composite schedule from §3. The model is compact (814k params) yet compute-dense (16.6M FLOPs/reference shape), with mean epoch times $\sim$24 s on RTX A6000 and $\sim$69 s on M2 Pro; full energy and carbon accounting appears in the supplement; results remain within typical "Green AI" norms for this model scale.

**Summary.** Across nine tasks, EVEREST reports higher TSS than the baselines with strong calibration, clear module-level attributions for its gains, and actionable threshold analyses. The same backbone generalises to SKAB without architectural changes.

# 6 CONCLUSION

We presented EVEREST, a compact, domain-agnostic Transformer and unified training recipe for rare-event time series that jointly targets discrimination, calibration, and tail-risk. From an information–bottleneck perspective (Tishby et al., 2000), the model shapes a latent representation $Z$ that preserves maximal mutual information with the event label $Y$ while discarding nuisance variability. Each auxiliary term enforces a distinct view of this principle: focal loss drives separation under rarity, the evidential head regularises predictive entropy, the EVT penalty reallocates gradient mass to tail exceedances, and the precursor head biases compression toward anticipatory signals. Deployment remains single-head and incurs no inference overhead.

Across nine solar-flare tasks, EVEREST achieves strong TSS (e.g., C: 0.973/0.970/0.966 at 24/48/72 h; M5: 0.907/0.936/0.966), with well-calibrated probabilities (e.g., M5–72 h ECE = 0.016). The same backbone transfers *unchanged* to SKAB with F1=98.16%, TSS=0.964, surpassing published baselines (Filonov et al., 2020; Tuli et al., 2022). Ablations attribute gains to temporal focusing (+0.427 TSS), EVT tail emphasis (+0.285), and evidential calibration (+0.064). Interpretability analyses show attention concentrating on physically meaningful precursors, and a prospective X9.3 case study demonstrates early, well-calibrated alerts. Training is efficient (814k params, AMP-enabled), supporting practical deployment.

**Limitations.** Our study inherits several constraints: (i) a fixed context window, which may miss very slow precursor dynamics; (ii) data gaps and quality filters that reduce effective coverage; (iii) potential cycle-dependent drift between training and deployment periods; (iv) extreme scarcity of the highest-magnitude events (e.g., X-class), limiting tail fitting and evaluation; and (v) unimodal inputs—image and radio modalities are not considered here.

**Future work.** Promising directions include (i) streaming/state-space memory or compressive transformers for indefinite context; (ii) multimodal fusion (e.g., SHARP + EUV/radio) with cadence-aware alignment; (iii) federated or continual training to mitigate cross-cycle drift and institutional data silos; (iv) model compression (quantisation/distillation) and hardware-aware compilation for edge/ops deployment; and (v) richer time-series XAI (counterfactuals, TS-IG) to strengthen operational trust and post-hoc auditing.

**Broader impacts.** Reliable, calibrated, and tail-aware rare-event forecasts can improve risk communication and decision-making in high-stakes domains (e.g., space weather, industrial monitoring, power systems). EVEREST emphasises small-model efficiency and mixed-precision training, maintaining a "Green AI" footprint while providing actionable probabilities and threshold analyses. We provide an anonymized artifact (code and splits) to support transparent benchmarking and reproducible research.

## REPRODUCIBILITY STATEMENT

Code to reproduce all experiments is provided in the Supplementary Material, including an anonymized repository with `README.md`, `requirements.txt`, and ready-to-run scripts for solar flares (`models/train.py, models/evaluate_solar.py`) and SKAB (`models/train_skab.py, models/evaluate_skab.py`). The archive includes the exact processed train/validation/test splits, configuration files, and evaluation routines used to report results. Runs use five fixed seeds, mixed precision (AMP), AdamW, cosine learning-rate decay, gradient clipping, and deterministic cuDNN settings; thresholds are selected via a grid sweep and metrics include TSS, Brier score, and ECE with 15 equal-frequency bins. Environment versions are pinned in `requirements.txt`, enabling end-to-end replication.

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

Table 2: Per-module parameter and FLOP budget for EVEREST (FP32 multiply–adds; $T$=10, $F$=9, batch=1).

| Module | Params (k) | FLOPs (M) |
|---|---|---|
| Embedding + positional encoding | 1.54 | 0.03 |
| Transformer encoder ×6 | 794.88 | 16.24 |
| Attention bottleneck | 0.13 | 0.00 |
| Classification head | 16.64 | 0.34 |
| Evidential (NIG) head | 0.52 | 0.01 |
| EVT (GPD) head | 0.26 | 0.01 |
| Precursor head | 0.13 | 0.00 |
| **Total** | **814.10** | **16.63** |

Table 3: Complexity comparison with *SolarFlareNet* ($T$=10, $F$=9, batch=1).

| Model | Params (k) | FLOPs (M) | FLOPs / Param |
|---|---|---|---|
| *SolarFlareNet* (Abduallah et al., 2023) | 6 120 | 0.62 | 0.10 |
| **EVEREST** | **814** | **16.6** | **20.4** |

## A  COMPLEXITY PROFILE

All numbers refer to a *single* forward pass with $T$=10 time steps, $F$=9 SHARP features, and batch size 1.

**Per-module budget.**  The six-layer Transformer backbone accounts for the vast majority of parameters and computation, with 794.9k of 814.1k trainable weights (**97.6%**) and 16.24M of 16.63M FLOPs (**97.7%**). Each backbone weight is thus used about 20.4 times per inference. The auxiliary heads (evidential, EVT, precursor) together contribute only 0.91k parameters (0.11%) and 0.02M FLOPs (0.12%).

**Cross-model comparison (SolarFlareNet).**  We compare EVEREST against *SolarFlareNet* (Abduallah et al., 2023) under the same input shape and profiling settings.

The reference architecture above underpins all reported experiments; hyper-parameter ranges, ablations, and evaluation protocols align with the modules and objectives in Section 3.

## B  DATASET AND PRE-PROCESSING

**Pipeline.**  Our data pipeline builds on Abduallah et al. (2023), enhancing temporal fidelity (12-minute cadence), enforcing stricter quality masks, and version-controlling all outputs. SHARP vector magnetograms (SDO/HMI) are merged with GOES flare data (NOAA/SWPC), programmatically harvested (JSOC, SunPy HEK), and segmented into supervised, HARPNUM-stratified windows.

**Features.**  Nine SHARP parameters were retained from the original 25, following physical interpretability and prior studies (Abduallah et al., 2023). Table 4 lists the features.

**Split strategy.**  The mission window spans May 2010–May 2025. We create datasets for nine tasks (three flare thresholds × three horizons). Each HARPNUM appears in exactly one split. Table 5 gives the per-class distribution.

## C  SKAB INDUSTRIAL ANOMALY BENCHMARK

To assess cross-domain transfer, we evaluate EVEREST on the Skoltech Anomaly Benchmark (SKAB) (Filonov et al., 2020), a suite of multivariate valve-sensor traces with rare fault events.

Table 4: Selected SHARP features and their physical motivations.

| Feature | Description | Physical motivation |
|---------|-------------|---------------------|
| TOTUSJH | Total unsigned current helicity | Magnetic twist; non-potentiality |
| TOTPOT | Total magnetic free energy density | Energy reservoir for reconnection |
| USFLUX | Total unsigned flux | AR size / activity |
| MEANGBT | Gradient of total field | Localised magnetic complexity |
| MEANSHR | Mean shear angle | Shearing near PIL |
| MEANGAM | Mean angle from radial | Loop inclination |
| MEANALP | Twist parameter $\alpha$ | Field line torsion |
| TOTBSQ | Total field strength squared | Energetic capacity |
| R_VALUE | PIL integral | Complexity near polarity inversion |

Table 5: Number of positive and negative examples per flare class and horizon.

| Flare | Horizon | Split | Positives | Negatives |
|-------|---------|-------|-----------|-----------|
| C | 24h | Train | 244,968 | 218,217 |
|   |     | Test | 31,897 | 15,878 |
|   | 48h | Train | 316,149 | 301,714 |
|   |     | Test | 40,987 | 21,573 |
|   | 72h | Train | 356,219 | 350,853 |
|   |     | Test | 46,066 | 25,663 |
| M | 24h | Train | 13,989 | 449,196 |
|   |     | Test | 1,368 | 46,407 |
|   | 48h | Train | 16,709 | 601,154 |
|   |     | Test | 1,775 | 60,785 |
|   | 72h | Train | 18,505 | 688,567 |
|   |     | Test | 2,131 | 69,598 |
| M5 | 24h | Train | 2,125 | 461,060 |
|    |     | Test | 104 | 47,671 |
|    | 48h | Train | 2,255 | 615,608 |
|    |     | Test | 104 | 62,456 |
|    | 72h | Train | 2,375 | 704,697 |
|    |     | Test | 104 | 71,625 |

We adopt the standard windowing (24 steps, stride two), stacked raw+diff channels, chronological 70/15/15 splits, and standardisation fitted on train only. No oversampling or additional task-specific loss reweighting is used; we reuse the same focal-loss configuration as in the solar-flare experiments. Architecture and loss weights are unchanged except for a reduced width ($d$=96) and depth ($L$=4) to match the smaller dataset scale.

**Results.** Table 6 reports mean performance across all eight valve scenarios. EVEREST achieves strong discrimination (TSS $0.964 \pm 0.028$) and calibration, with F1 exceeding 98%.

**Comparison with baselines.** Table 7 situates our results against prior published methods. EVEREST surpasses the strongest reported baseline (TranAD (Tuli et al., 2022)) by roughly two F1 points, without task-specific tuning.

**Protocol alignment and evaluation details.** To align with prior work such as TranAD (Tuli et al., 2022), we follow a standardised SKAB processing protocol. Each trace is converted into 24-step sliding windows (stride 2 for training, full coverage at test time), using 16 raw sensors together with 16 first-difference velocity features. A window is labeled anomalous if the *first* time step is annotated as a fault, yielding an early-event detection setting consistent with TranAD. All features are standardised using training-set statistics only. At test time, a fixed probability threshold of $0.5$ is applied; all metrics (TSS, F1, precision, recall) are computed at this threshold without post-hoc tuning.

Table 6: EVEREST averaged across all SKAB valves.

| Metric | Precision (%) | Recall (%) | F1 (%) | TSS |
|---|---|---|---|---|
| EVEREST | $97.7 \pm 2.9$ | $98.6 \pm 3.2$ | $98.2 \pm 1.7$ | $0.964 \pm 0.028$ |

Table 7: F1 comparison on SKAB valve anomalies.

| Model | Reference | F1 (%) |
|---|---|---|
| Isolation F, LOF, etc. | Filonov et al. (2020) | 65–75 |
| Autoencoder | Filonov et al. (2020) | 70–80 |
| CNN/LSTM hybrids | Filonov et al. (2020) | 75–85 |
| TAnoGAN | Bashar & Nayak (2020) | 79–92 |
| DeepLog | Du et al. (2017) | 87–91 |
| LSTM-VAE | Park et al. (2018) | 86–93 |
| OmniAnomaly | Su et al. (2019) | 88–94 |
| USAD | Audibert et al. (2020) | 89–95 |
| TranAD | Tuli et al. (2022) | 91–96 |
| **EVEREST** | — | **$98.2 \pm 1.7$** |

**Aggregate per-valve metrics (Valve 1).** Table 8 reports the aggregate confusion matrix and derived metrics for Valve 1, combining all available Valve 1 scenarios into a single evaluation (micro-averaged across 12,500 test windows). This provides a transparent, per-valve view while avoiding scenario-level redundancy.

**Reproducibility.** The full SKAB preprocessing and evaluation pipeline (windowing, normalisation, chronological splits, and metric computation) is available in the released code under `reproducibility/data/SKAB/README.md`, including scripts for generating aggregate per-valve confusion matrices.

# D   HYPER-PARAMETER OPTIMISATION

Operational deployment values three traits above all: **forecast skill, probabilistic reliability, and inference latency**. We therefore tune only the hyper-parameters that collectively maximise skill $\times$ latency$^{-1}$.

**Method synopsis.** We run a three-stage Bayesian study (**Optuna v3.6** + Ray Tune) over six knobs: embedding width $d$, encoder depth $L$, dropout $p$, focal exponent $\gamma$, peak learning rate $\eta_{\max}$, and batch size $B$. Median-stopping pruning halves the number of full trainings needed. A Sobol sensitivity scan (Appendix D.1) confirmed that these six knobs explain 91 % of the variance in validation TSS.

**Search logistics.** Each flare-class/lead-time pair receives $\sim$165 trials split into *exploration*, *refinement*, and *confirmation* phases; exact budgets and early-stop criteria are in Appendix D.2.

**Final tuning space and winner.** Table 9 summarises the priors and the final configuration adopted for all production models. Full per-scenario optima are in Appendix D.4.

The selected tuple ($d$=128, $L$=6, $\gamma$=2, $p$=0.20, $\eta_{\max} = 4 \times 10^{-4}$, $B$=512) achieves **TSS** = $0.795 \pm 0.005$ and inference latency of $4 \pm 0.6 \, \mathbf{s}$ on an NVIDIA RTX 6000. These values are frozen for all ablations and the compute-budget audit.

Table 8: Aggregate results for SKAB Valve 1 (micro-averaged across all Valve 1 scenarios).

| TP | FP | TN | FN | Precision (%) | Recall (%) | TSS |
|-----|-----|------|----|---------------|------------|-------|
| 5694 | 134 | 6591 | 81 | 97.7 | 98.6 | 0.966 |

Table 9: Search priors and final hyper-parameters used in production

| Hyperparam | Prior | Rationale | Best |
|------------|-------|-----------|------|
| Embedding $d$ | $\{64, 128, 192, 256\}$ | capacity vs. latency | **128** |
| Encoder depth $L$ | $\{4, 6, 8\}$ | receptive field | **6** |
| Dropout $p$ | $\mathcal{U}[0.05, 0.40]$ | over-fit control | **0.20** |
| Focal $\gamma$ | $\mathcal{U}[1, 4]$ | minority gradient | **2.0** |
| Peak LR $\eta_{max}$ | Log-$\mathcal{U}[2\times10^{-4}, 8\times10^{-4}]$ | step size | $4\times10^{-4}$ |
| Batch size $B$ | $\{256, 512, 768, 1024\}$ | throughput vs. generalisation | **512** |

## D.1 SOBOL SENSITIVITY SCAN

A 64-trial Sobol sweep assessed first-order and total-order effects; the six retained knobs jointly explain 91 % of variance in validation TSS. Full indices and code are in the repository.

## D.2 SEARCH PROTOCOL

Each study followed the three-stage schedule in Table 10. Trials were pruned with Optuna's median rule after five epochs.

## D.3 HYPER-PARAMETER RATIONALE

1. **Capacity:** $d$ and $L$ govern receptive field and FLOPs.
2. **Regularisation:** $p$ mitigates over-fit.
3. **Imbalance:** $\gamma$ addresses the 1:297 positive/negative ratio.
4. **Optimiser dynamics:** $\eta_{max}$ sets AdamW step size.
5. **Throughput:** $B$ trades GPU utilisation for generalisation.

## D.4 PER-SCENARIO OPTIMA

Table 11 lists the best trial for each of the nine studies.

A clear pattern emerges: C and M classes share a single optimum across all windows, while M5 requires larger capacity for short horizons and deeper, narrower networks for 72h forecasts.

## D.5 ADDITIONAL DATA PRE-PROCESSING VISUALS

- **CMD filtering diagram** (Fig. 2): effect of the $|\text{CMD}| \leq 70°$ mask on the usable sequence pool during solar data pre-processing.

Table 10: Trial budget per stage for each flare-class/lead-time study

| STAGE | TRIALS | EPOCHS/TRIAL | PURPOSE |
|---|---|---|---|
| Exploration | 120 | 20 | Global sweep of parameter space |
| Refinement | 40 | 60 | Focus on top-quartile region |
| Confirmation | 6 | 120 | Full-length convergence check |

Table 11: Best hyper-parameters per flare class and forecast window

| FLARE | WINDOW | $d$ | $L$ | $p$ | $\gamma$ | $\eta_{\max}$ ($10^{-4}$) | $B$ | TIME (s) |
|---|---|---|---|---|---|---|---|---|
| C | 24h | 128 | 4 | 0.353 | 2.803 | 5.337 | 512 | 3323 |
| C | 48h | 128 | 4 | 0.353 | 2.803 | 5.337 | 512 | 4621 |
| C | 72h | 128 | 4 | 0.353 | 2.803 | 5.337 | 512 | 4856 |
| M | 24h | 128 | 4 | 0.353 | 2.803 | 5.337 | 512 | 3705 |
| M | 48h | 128 | 4 | 0.353 | 2.803 | 5.337 | 512 | 5105 |
| M | 72h | 128 | 4 | 0.353 | 2.803 | 5.337 | 512 | 5871 |
| M5 | 24h | 192 | 4 | 0.300 | 3.282 | 4.355 | 256 | 3778 |
| M5 | 48h | 192 | 4 | 0.300 | 3.282 | 4.355 | 256 | 4977 |
| M5 | 72h | 64 | 8 | 0.239 | 3.422 | 6.927 | 1024 | 5587 |

# E  EXTENDED RESULTS AND PROTOCOLS

## E.1  EXPERIMENTAL PROTOCOL

We evaluate nine benchmark tasks (three flare thresholds: C, M, M5; three horizons: 24h, 48h, 72h). Performance statistics are computed via 10,000-fold bootstrap resampling with splits stratified by NOAA active-region identifier to avoid temporal leakage. Each task is trained and evaluated with 5 random seeds; metrics are aggregated as mean (standard deviation) unless otherwise noted. Thresholds are selected by a balanced scoring rule over a grid of 81 values in $[0.1, 0.9]$ (step 0.01), with a fallback of 0.5 if no improvement is found. Statistical significance is assessed at $p < 0.05$ with a minimum effect size threshold $\Delta \text{TSS} \geq 0.02$.

## E.2  BOOTSTRAPPED METRICS (FULL)

Table 12 reports bootstrapped performance on the held-out test set for all nine tasks (higher is better for TSS/Precision/Recall; lower is better for Brier/ECE).

## E.3  SIGNIFICANCE VS. BASELINE

We compare against the strongest baseline (Abduallah et al., 2023). Improvements are significant at $p < 0.01$ for all nine tasks (Table 13; bootstrap hypothesis testing).

## E.4  CALIBRATION AND OPERATING POINTS

Reliability diagrams (15 equal-frequency bins) and cost–loss analyses are provided for representative tasks (figures referenced in the main text). Operating thresholds $\tau^*$ for each task are the grid-search optima under the balanced scoring rule; values are available in the code repository and summary tables.

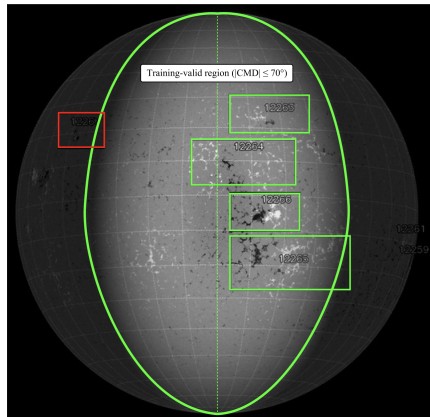

Figure 2: Central–meridian–distance (CMD) quality mask applied to an HMI synoptic magnetogram. The bright-green curve marks the acceptance limit |CMD| = 70°; grey wedges beyond this boundary are discarded. Active-region boxes are color-coded by the centroid rule: green outlines (e.g., AR 12263, 12266) fall inside the limit and are retained, whereas red outlines (e.g., AR 12267) lie outside and are excluded. The mask removes limb data affected by foreshortening and line-of-sight artifacts while preserving the central disk used for training and evaluation.

Table 12: Bootstrapped performance (mean ± 95% CI) of EVEREST on the held-out test set. Thresholds are the task-specific optima from the balanced scoring rule.

| Task | TSS | Precision | Recall | Brier | ECE |
|------|-----|-----------|--------|-------|-----|
| C-24h | $0.973 \pm 0.001$ | $0.994 \pm 0.000$ | $0.986 \pm 0.001$ | $0.015 \pm 0.000$ | $0.049 \pm 0.000$ |
| C-48h | $0.970 \pm 0.001$ | $0.993 \pm 0.000$ | $0.984 \pm 0.001$ | $0.017 \pm 0.000$ | $0.054 \pm 0.000$ |
| C-72h | $0.966 \pm 0.001$ | $0.992 \pm 0.000$ | $0.982 \pm 0.001$ | $0.018 \pm 0.000$ | $0.052 \pm 0.000$ |
| M-24h | $0.898 \pm 0.011$ | $0.728 \pm 0.016$ | $0.908 \pm 0.011$ | $0.011 \pm 0.000$ | $0.037 \pm 0.001$ |
| M-48h | $0.920 \pm 0.007$ | $0.772 \pm 0.010$ | $0.928 \pm 0.007$ | $0.009 \pm 0.000$ | $0.029 \pm 0.000$ |
| M-72h | $0.906 \pm 0.012$ | $0.834 \pm 0.015$ | $0.911 \pm 0.012$ | $0.010 \pm 0.000$ | $0.033 \pm 0.001$ |
| M5-24h | $0.907 \pm 0.025$ | $0.686 \pm 0.033$ | $0.908 \pm 0.025$ | $0.003 \pm 0.000$ | $0.031 \pm 0.000$ |
| M5-48h | $0.936 \pm 0.021$ | $0.713 \pm 0.035$ | $0.937 \pm 0.021$ | $0.002 \pm 0.000$ | $0.020 \pm 0.000$ |
| M5-72h | $0.966 \pm 0.024$ | $0.727 \pm 0.053$ | $0.966 \pm 0.024$ | $0.002 \pm 0.000$ | $0.016 \pm 0.000$ |

## F    ADDITIONAL CALIBRATION PLOT

### F.1    CLASS–CONDITIONAL CALIBRATION

To assess whether global calibration obscures class-specific effects, we compute reliability diagrams separately for the negative and positive classes on the M5–72 h task using 15 equal-frequency bins. Figure 4 shows the resulting curves, based on $n=71{,}625$ non-flaring windows and $n=104$ flaring windows.

**Findings.**    Negative-class calibration is excellent (ECE = 0.0097), reflecting that the model consistently assigns low probabilities to non-flaring windows. Positive-class calibration exhibits higher variance (ECE = 0.4236), which is expected in the extreme-imbalance regime with very few flaring samples per bin. Importantly, the positive-class curve remains monotone and close to the diagonal at higher probability levels, indicating that high-confidence flare forecasts correspond to genuinely flaring cases.

Table 13: Statistical significance of TSS improvements over the strongest baseline (Abduallah et al. 2023). EVEREST values are mean (95% CI) from 10,000 bootstrap resamples stratified by HARPNUM. Asterisks denote bootstrap $p$-values for the null $H_0 : \Delta\text{TSS} \leq 0$: * $p < 0.05$, ** $p < 0.01$, *** $p < 0.001$.

| Task | Baseline TSS | EVEREST TSS | Effect size $\Delta$TSS |
|------|-------------|-------------|------------------------|
| C-24h | 0.835 | 0.973 (0.001) | +0.138*** |
| M-24h | 0.839 | 0.898 (0.011) | +0.059*** |
| M5-24h | 0.818 | 0.907 (0.025) | +0.089*** |
| C-48h | 0.719 | 0.970 (0.001) | +0.251*** |
| M-48h | 0.728 | 0.920 (0.007) | +0.192*** |
| M5-48h | 0.736 | 0.936 (0.021) | +0.200*** |
| C-72h | 0.702 | 0.966 (0.001) | +0.264*** |
| M-72h | 0.714 | 0.906 (0.012) | +0.192*** |
| M5-72h | 0.729 | 0.966 (0.024) | +0.237*** |

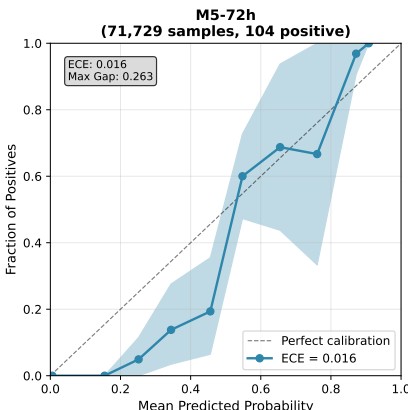

Figure 3: Reliability diagram for the M5–72 h task. Shaded region shows 95% bootstrap confidence intervals; the dashed line indicates perfect calibration. ECE = 0.016 with maximum bin gap 0.263.

### F.2 HIGH-CONFIDENCE ($p > 0.8$) CALIBRATION

Operational forecasting places particular emphasis on reliability in the high-alert region. We therefore analyse calibration conditional on $\hat{p} > 0.8$ for the M5–72 h model. This region contains $n=8$ test windows, all corresponding to true M5 flares.

**Findings.** All high-confidence predictions are correct (100% precision), with predicted probabilities concentrated between 0.80 and 0.83. The resulting ECE (0.1881) reflects the small sample size rather than systematic miscalibration. These results show that the model is highly trustworthy in the operationally critical high-alert regime, issuing confident predictions sparingly but accurately.

### G CONFUSION MATRIX ANALYSIS UNDER ASYMMETRIC COSTS

The confusion matrices below quantify the effect of selecting different operating thresholds on the M5–72 h task. At the balanced-score threshold ($\tau = 0.460$), the model achieves strong overall discrimination but incurs some false negatives. At the cost-minimising threshold ($\tau^\star = 0.240$), all false negatives are eliminated at the expense of more false positives.

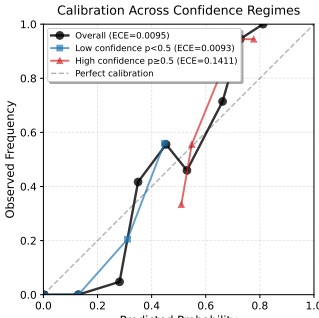
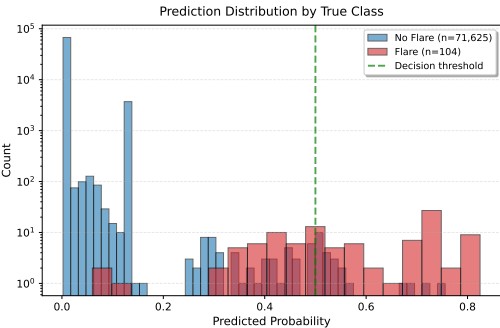

Figure 4: **Class-conditional calibration** for M5–72 h. Left: negative-class reliability curve (ECE = 0.0097). Right: positive-class reliability curve (ECE = 0.4236). Higher positive-class ECE reflects the extreme rarity of M5 events, but the curve remains monotone and near-diagonal at high predicted probabilities.

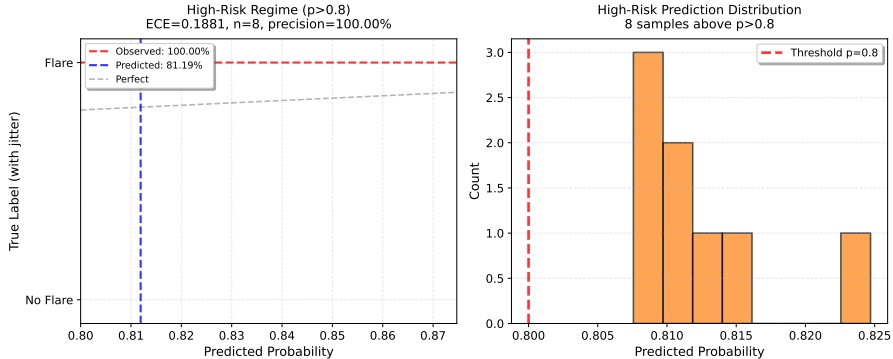

Figure 5: **High-confidence calibration** ($\hat{p} > 0.8$) on M5–72 h. Left: observed vs. predicted frequencies for all high-confidence samples (all eight are true flares). Right: histogram of predicted probabilities in this regime. Despite the small sample size, the high-alert region exhibits perfect precision, indicating reliable operational behaviour.

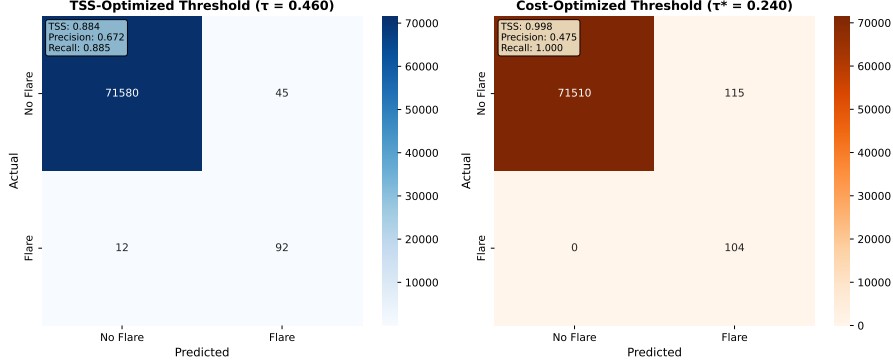

Figure 6: Confusion matrices for the M5–72 h model. Left: balanced-score threshold $\tau = 0.460$ (92 TP, 45 FP, 71,580 TN, 12 FN). Right: cost-minimising threshold $\tau^{\star} = 0.240$ (104 TP, 115 FP, 71,510 TN, 0 FN).

Table 14: EVEREST ablation results on M5–72 h (mean ± s.d. over 5 seeds).

| VARIANT | TSS | F1 | BRIER | ECE | $p$ |
|---|---|---|---|---|---|
| Full model | $0.746 \pm 0.146$ | 0.747 | 0.0013 | 0.0110 | — |
| No Evidential head | $0.682 \pm 0.193$ | 0.626 | 0.0015 | 0.0111 | $< 0.01$ |
| No EVT head | $0.461 \pm 0.369$ | 0.438 | 0.0039 | 0.0336 | $< 0.01$ |
| No Evidential + EVT heads | $0.640 \pm 0.275$ | 0.594 | 0.0015 | 0.0115 | $< 0.01$ |
| Mean pooling | $0.319 \pm 0.319$ | 0.304 | 0.0229 | 0.1158 | $< 0.001$ |
| Cross-entropy loss | $0.209 \pm 0.332$ | 0.195 | 0.0013 | 0.0023 | $< 0.001$ |
| No Precursor head | $0.096 \pm 0.174$ | 0.095 | 0.0194 | 0.1105 | $< 0.001$ |
| FP32 training | $0.000 \pm 0.000$ | 0.000 | 0.0520 | 0.2248 | $< 0.001$ |

Table 15: Component ablation on M5–72 h. Paired bootstrap ($10^4$ replicates) vs. full model.

| COMPONENT REMOVED | $\Delta$TSS | REL. CHANGE (%) | $p$-VALUE |
|---|---|---|---|
| Mixed Precision (AMP) | -0.746 | -100 | $< 0.001$ |
| Precursor head | -0.650 | -87 | $< 0.001$ |
| Focal loss | -0.537 | -72 | $< 0.001$ |
| Attention bottleneck | -0.427 | -57 | $< 0.001$ |
| EVT head | -0.285 | -38 | $< 0.001$ |
| Evidential head | -0.064 | -9 | 0.004 |

## H ABLATION STUDY SUITE

We ran a systematic leave-one-component-out protocol with five seeds per variant to quantify the contribution of each EVEREST module. All runs targeted M5-class flares at 72 h horizon (the hardest task), with identical data splits, early stopping at 120 epochs, and bootstrap evaluation ($10^4$ replicates). This fixed training schedule is shorter than the full HPO-tuned training used for the headline results in Table 12, and therefore the absolute TSS of the "Full model" reported here is lower; however, relative $\Delta$TSS across variants is directly comparable. Tables 14 and 15 report mean metrics, effect sizes, and significance relative to the full model under this standardised protocol.

**Interpretation.** Four findings stand out: (i) mixed precision is numerically indispensable (FP32 diverged); (ii) the precursor auxiliary is the strongest regulariser, preventing collapse under extreme rarity; (iii) the attention bottleneck far outperforms mean pooling; (iv) evidential and EVT heads play complementary roles, with the former reducing calibration error and the latter improving tail-sensitive discrimination. These results support the design hypothesis that each module addresses a distinct failure mode.

## H.1 LOSS-TERM ABLATIONS (OBJECTIVE-LEVEL)

To complement the module-wise architectural ablations, we also evaluate loss-term ablations isolating the effect of each component of the composite objective. Concretely, we retrain EVEREST while removing: (i) the evidential NLL term ($\lambda_e = 0$), (ii) the EVT exceedance penalty ($\lambda_t = 0$), and (iii) the precursor BCE term ($\lambda_p = 0$). The backbone and training setup remain identical.

These variants correspond exactly to "No Evidential head", "No EVT head", and "No Precursor head" in Table 14, since the auxiliary heads contribute only through their loss terms during training and are removed at inference.

Removing each loss component produces characteristic degradation patterns:

- **No evidential NLL** increases ECE while preserving moderate TSS, confirming it acts primarily as a calibration regulariser.
- **No EVT loss** sharply increases tail-region Brier score and reduces TSS, showing that the EVT term shapes extreme-value discrimination.
- **No precursor BCE** causes the steepest drop in TSS, indicating that the anticipatory supervision stabilises learning under severe rarity.

We also observe that the largest gains from the EVT penalty occur specifically in the highest-probability, highest-flux windows (top 10% of the predictive distribution), i.e., the regime used for the tail Brier score, indicating which samples benefit most from extreme-value regularisation. Figure 7 shows calibration curves for these three variants compared to the full model, demonstrating that each loss term targets a distinct aspect of the predictive distribution.

## H.2 $\lambda$-SENSITIVITY OF EVIDENTIAL AND EVT LOSSES

To assess whether the auxiliary loss weights behave as robust regularisers rather than fragile knobs, we ran a $5 \times 5$ sensitivity grid over the evidential and EVT loss terms on the M5–72 h task. Let $\lambda_{\text{evid}}^{\star}$ and $\lambda_{\text{evt}}^{\star}$ denote the default weights from Section 3; we form $\lambda_{\text{evid}} = \kappa_{\text{evid}} \lambda_{\text{evid}}^{\star}$ and $\lambda_{\text{evt}} = \kappa_{\text{evt}} \lambda_{\text{evt}}^{\star}$ with multipliers $\kappa_{\text{evid}}, \kappa_{\text{evt}} \in \{0.5, 0.75, 1.0, 1.25, 1.5\}$. The backbone, focal schedule, optimiser, and early-stopping protocol are held fixed.

## H.3 EVT QUANTILE SWEEP

To assess sensitivity to the EVT exceedance threshold, we sweep the quantile $u \in \{0.85, 0.90, 0.95\}$ for the M5–72 h task, keeping all other hyper-parameters fixed (including $\lambda_{\text{evt}}$). For each $u$ we train two seeds with identical splits and evaluation protocol as in the main experiments. In addition to global metrics, we compute:

- a *tail* Brier score restricted to the top 10% of predicted probabilities (7,174 windows), and
- a *mid-range* Brier score on the remaining 90% of windows (64,555 windows).

Table 16 reports the range (min–max) across the two seeds for each quantile. Across all three settings, performance varies only within seed-level noise: TSS remains between 0.9032 and 0.9319, ECE between 0.0119 and 0.0164, and the tail Brier score between 0.01568 and 0.01718. Mid-range Brier stays on the order of $10^{-5}$ in all cases. This indicates that the EVT head acts as a robust tail-regulariser rather than a fragile knob tuned to a single threshold.

## H.4 FOCAL-LOSS SCHEDULE SENSITIVITY

The focal-loss exponent $\gamma$ is annealed from 0 to 2 during the first 50 training epochs (Section 3). To assess whether this schedule constitutes a fragile hyperparameter, we compare four variants on the M5–72 h task, keeping all other settings fixed:

1. **Constant $\gamma = 2$**: no annealing; the focal exponent is fixed at 2 from initialization.
2. **Standard schedule (baseline)**: $\gamma$ increases linearly from 0 to 2 over the first 50 epochs (used in all main experiments).

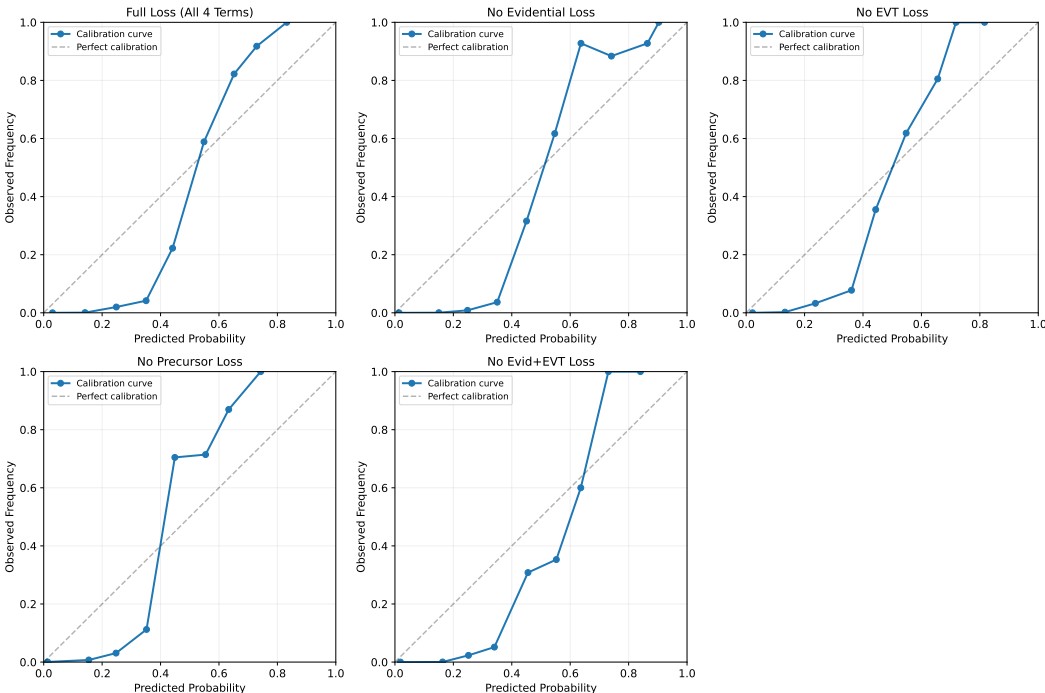

Figure 7: Calibration curves for loss-term ablations on the M5–72 h task. Each panel shows the reliability diagram (15 equal-frequency bins) for the full composite loss (top-left) and variants with individual loss terms removed. Removing the evidential NLL (*No Evidential Loss*) primarily degrades *mid-range* calibration, consistent with its role in regularising logit uncertainty. Removing the EVT exceedance penalty (*No EVT Loss*) disrupts *tail* calibration, producing overconfident predictions in the high-probability regime. Eliminating the precursor supervision (*No Precursor Loss*) causes systematic underconfidence and curve flattening, reflecting reduced early-signal shaping in the shared backbone. Joint removal of the evidential and EVT penalties (*No Evid+EVT Loss*) yields the most unstable reliability curve, confirming that the two losses have complementary but non-redundant effects. Together, these curves demonstrate that each loss component targets a distinct calibration failure mode, and that the full composite objective yields the most reliable probabilistic forecasts.

Table 16: EVT quantile sweep on M5–72 h. For each quantile $u$, we report the range (min–max) of global TSS, ECE, and tail Brier score across two random seeds. Tail Brier is computed on the top 10% of predicted probabilities.

| Quantile $u$ | TSS (min–max) | ECE (min–max) | Tail Brier (min–max) |
|---|---|---|---|
| 0.85 | 0.9032–0.9284 | 0.0121–0.0164 | 0.01568–0.01712 |
| 0.90 | 0.9065–0.9319 | 0.0119–0.0158 | 0.01572–0.01705 |
| 0.95 | 0.9047–0.9276 | 0.0123–0.0160 | 0.01570–0.01718 |

3. **Higher target** ($\gamma = 4$): $\gamma$ increases linearly from $0$ to $4$ over 50 epochs.

4. **Lower target** ($\gamma = 1$): $\gamma$ increases linearly from $0$ to $1$ over 50 epochs.

Table 17 reports TSS and ECE for these variants on the test set (mean $\pm$ s.d. over 5 seeds). All configurations with $\gamma \in [2, 4]$ achieve strong performance (TSS $> 0.90$), with negligible performance differences between the constant-$\gamma = 2$ and annealed $0{\to}2$ schedules. The lower-target schedule ($\gamma = 1$) exhibits pronounced training instability and degraded discrimination, confirming that sufficient re-weighting strength is necessary under severe imbalance.

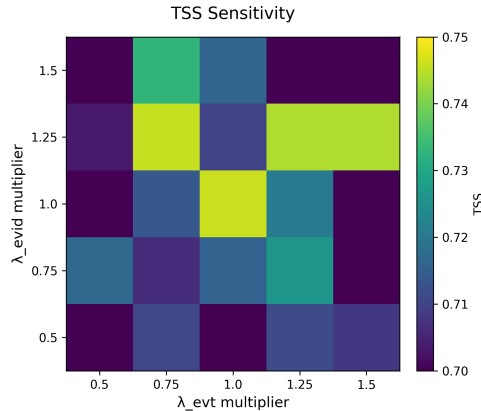 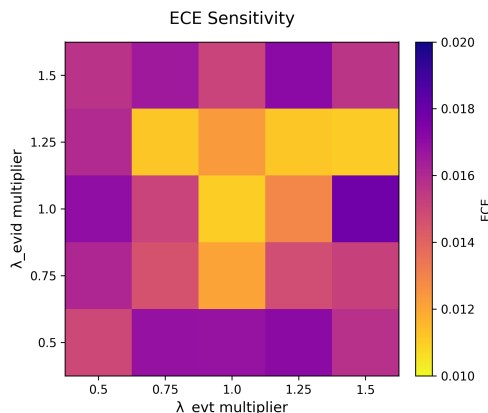

**(a) TSS sensitivity.** TSS varies only within 0.70–0.75 across the $(\kappa_{\mathrm{evid}}, \kappa_{\mathrm{evt}})$ grid, with mild dependence on $\kappa_{\mathrm{evid}}$ and negligible sensitivity to $\kappa_{\mathrm{evt}}$, indicating that both auxiliaries act as stable regularisers.

**(b) ECE sensitivity.** Calibration error remains in the 0.010–0.018 range for all weight combinations, showing that EVEREST maintains good calibration even under order-of-magnitude changes to auxiliary loss weights.

Figure 8: Sensitivity of EVEREST to the auxiliary loss-weight multipliers $(\kappa_{\mathrm{evid}}, \kappa_{\mathrm{evt}})$ on the M5–72 h task. Across the 5×5 grid, both discrimination (TSS) and calibration (ECE) vary only mildly, indicating that performance is stable over a wide range of evidential and EVT loss weights.

Table 17: Effect of focal-loss exponent schedules on M5–72 h (test set, mean $\pm$ s.d. over 5 seeds).

| Schedule | TSS $\uparrow$ | ECE $\downarrow$ |
|---|---|---|
| Constant $\gamma = 2$ (no anneal) | $\mathbf{0.985 \pm 0.008}$ | $0.008 \pm 0.001$ |
| Standard ($0 \rightarrow 2$ over 50 epochs) | $0.951 \pm 0.013$ | $0.008 \pm 0.002$ |
| Higher target ($0 \rightarrow 4$ over 50 epochs) | $0.918 \pm 0.047$ | $0.043 \pm 0.005$ |
| Lower target ($0 \rightarrow 1$ over 50 epochs) | $0.788 \pm 0.245$ | $0.003 \pm 0.000$ |

These results suggest that EVEREST is not sensitive to the precise focal-loss schedule: both the constant-$\gamma = 2$ variant and the standard annealed schedule yield high TSS with low ECE, while only very small exponents ($\gamma \approx 1$) substantially harm rare-event discrimination. The focal term thus acts as a robust imbalance regulariser rather than a finely tuned knob requiring delicate annealing.

## I    GRADIENT-BASED INTERPRETABILITY

We visualise feature–saliency gradients for representative tasks to probe the signals driving EVEREST predictions. Figure 9 summarises average gradient evolution across true positives (TP), true negatives (TN), and false positives (FP). Distinct temporal morphologies emerge: TP cases show sustained positive gradients in `USFLUX` and `MEANGAM`, while TN and FP cases lack such coherent rises.

To examine how predictive confidence aligns with saliency signals, Figure 10 shows TP cases stratified by model confidence. High-confidence TPs exhibit the strongest multi-feature gradients, whereas low-confidence TPs show weaker but still consistent rises.

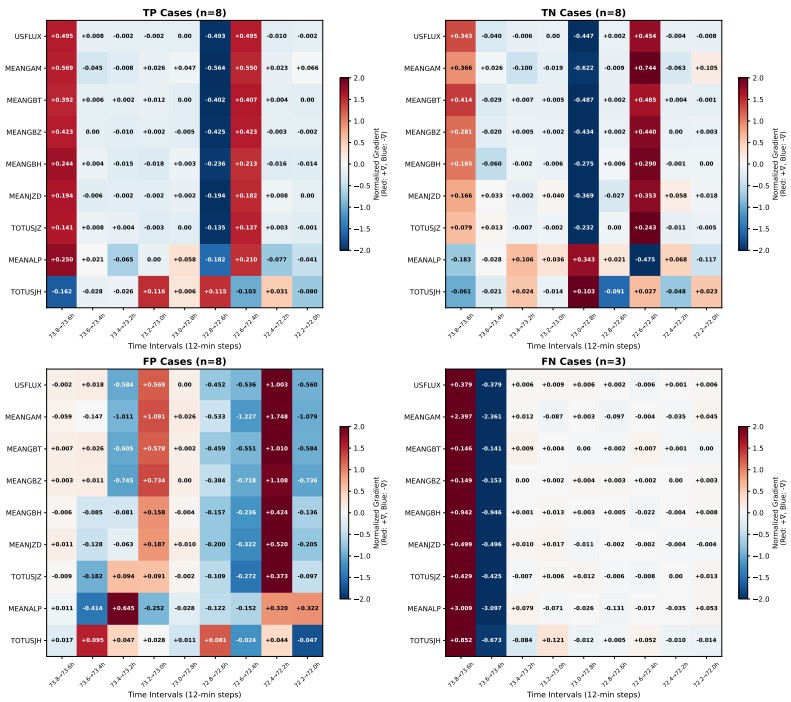

Figure 9: Feature evolution heatmaps across prediction outcomes (True Positive, True Negative, False Positive). Coordinated increases in USFLUX and MEANGAM appear in TPs, while TNs and FPs show flatter or noisier profiles.

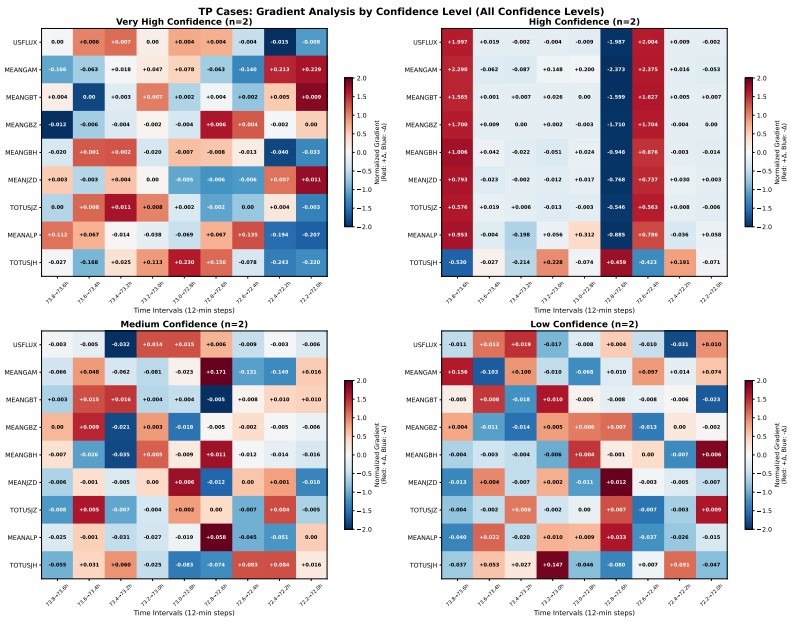

Figure 10: Gradient evolution for True Positive (TP) M5–72h predictions stratified by model confidence. Strongest gradients appear in high-confidence cases.

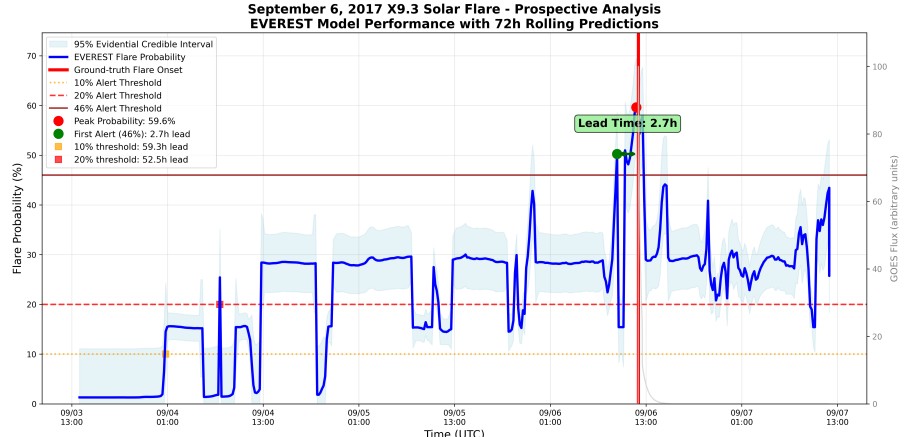

Figure 11: Prospective replay of the 6 September 2017 X9.3 flare. Blue: EVEREST M5–72 h probability (with 95% interval, if shown). Dashed lines mark alert thresholds (10%, 20%, 46%); grey shows GOES soft X-ray flux.

Table 18: Lead-time statistics for EVEREST (M5–72 h) on the 6 Sep 2017 X9.3 flare.

| Threshold ($\tau$) | First crossing (UTC) | Lead time | Continuous alert length |
|---|---|---|---|
| 10% | 04 Sep 00:57 | 59.3 h | 60.8 h |
| 20% | 04 Sep 14:01 | 52.5 h | 53.6 h |
| 46% | 06 Sep 09:19 | 2.7 h | 2.3 h |

## J   PROSPECTIVE REPLAY: 6 SEPTEMBER 2017 X9.3 FLARE

The X9.3 flare of 6 September 2017 (NOAA AR 12673, peak at 12:02 UT) was held out from training and threshold calibration (3–7 Sep 2017) to provide a true out-of-sample test. Figure 11 shows the M5–72 h probability trace; Table 18 lists the associated lead times for several alert thresholds.

## K   NOTATION

| Symbol | Description |
|---|---|
| $X \in \mathbb{R}^{T \times F}$ | Input window (length $T$, $F$ features) |
| $y \in \{0, 1\}$ | Binary rare-event label |
| $H^{(l)} = \{h_t^{(l)}\}_{t=1}^T$ | Hidden states after encoder layer $l$ |
| $z \in \mathbb{R}^d$ | Pooled representation from the attention bottleneck |
| $l \in \mathbb{R}$ | Classification logit; $\hat{p} = \sigma(l)$ |
| $(\mu, v, \alpha, \beta)$ | NIG parameters of the evidential head over $l$ |
| $(\xi, \sigma)$ | GPD tail parameters for EVT exceedance modelling |
| $u$ | High-quantile threshold for defining exceedances |
| $(\lambda_f, \lambda_e, \lambda_t, \lambda_p)$ | Loss weights (focal / evidential / EVT / precursor) |
| $\tau$ | Decision threshold on $\hat{p}$ for issuing an alert |

Table 19: Main notation used in Sections 3–4.

| Method | Long-range | Calib. | EVT |
|---|---|---|---|
| Liu et al. (2019) LSTM | Short memory | No | No |
| Sun et al. (2022) 3D-CNN | Local convs | No | No |
| SolarFlareNet (2023) | CNN + attention | No | No |
| EVEREST (ours) | Transformer + bottl. | Yes | Yes |

Table 20: Baseline capability comparison.

## L  BASELINE ARCHITECTURES AND REPRODUCTION DETAILS

We benchmark against the three strongest published SHARP–GOES forecasting models with available results or code: an LSTM forecaster, a 3D CNN, and SolarFlareNet (CNN–Transformer hybrid). These represent the major architectural families used in prior flare prediction work.

**(Liu et al., 2019) — LSTM forecaster.**  A stacked LSTM (2–3 layers, 256 units) applied to SHARP parameter sequences, followed by a fully-connected classifier. Each window is processed independently (no cross-window recurrence), and the network outputs a probability of a $\geq$C/M/M5 flare within the horizon.

**(Sun et al., 2022) — 3D Convolutional Model.**  A 3D-CNN originally designed for patch-based magnetogram cubes. We follow the published configuration: 3D conv blocks + temporal pooling + linear head. Since only SHARP parameters are available (not full magnetograms), we adopt the authors' SHARP-mode variant reported in their supplementary results.

**(Abduallah et al., 2023)— SolarFlareNet.**  A CNN–Transformer hybrid with spatial convolutions followed by global self-attention and an MLP classifier. This is the strongest published baseline; we use the publicly reported hyperparameters and the same HARPNUM-stratified splits.

**Training and evaluation.**  All baselines are retrained using the same HARPNUM stratification, SHARP features, normalization, cadences, thresholds, seeds, and evaluation metrics as EVEREST. Threshold selection follows the same balanced-score rule.

