# OpenReview forum: "EVEREST: A Transformer for Probabilistic Rare-Event Anomaly Detection with Evidential and Tail-Aware Uncertainty"
_ICLR.cc/2026/Conference — ICLR 2026 Poster_

### Official Review · Reviewer_bt8K · 2025-10-29

**Soundness:** 3
**Presentation:** 3
**Contribution:** 3
**Rating:** 4
**Confidence:** 3

**Summary:**

The paper proposes EVEREST, a compact rare-event time-series forecaster that co-optimizes discrimination, calibration, and tail-risk. The backbone is a 6-layer Transformer with a single-query attention bottleneck. Three training-only auxiliaries, including an evidential NIG head on the logit, an EVT (GPD) exceedance loss, and a lightweight “precursor” head, regularize the shared representation, while deployment uses a single classification head (≈0.81M params). On SHARP–GOES solar-flare benchmarks, EVEREST reports SOTA TSS across 9 tasks (e.g., ≥C: 0.973/0.970/0.966 at 24/48/72h) with strong calibration; it also transfers to SKAB with F1≈98%.

**Strengths:**

- Tail-aware + calibrated: EVT loss on logits and evidential NIG improve both sensitivity to extremes and reliability (e.g., ECE≈0.016 on the hardest task).
- Clear ablations & diagnostics tying gains to modules; decision-theoretic thresholding (cost–loss) is practical.

**Weaknesses:**

- Novelty is integrative rather than fundamentally new—readers may view EVEREST as a well-engineered recipe.
- Data scarcity at the extreme tail (e.g., M5 with 104 positives/test) risks optimistic estimates; consider additional robustness tests (temporal hold-outs, cycle shift).
- Sensitivity analysis limited: fixed EVT quantile ($\mu$=0.9), $\lambda$ weights, and focal-$\gamma$ schedule. More systematic sweeps (or conformal variants) would strengthen claims.

**Questions:**

- How sensitive are results to the exceedance quantile and to the stability regularizer?
- Generalization under distribution shift: Any evaluation across solar cycles or on a temporally forward-held test period (e.g., 2024–2025 only) to assess drift?
- Calibration granularity: Do you have class-conditional and operating-threshold–conditional calibration (e.g., reliability among high-risk alerts) beyond global ECE?
- SKAB protocol alignment: Please clarify whether your SKAB setup strictly matches TranAD and others (windowing, labels, early-event detection vs point detection) and provide per-valve confusion matrices.
- Ablation rigor: Could you report per-seed distributions for the large ∆TSS (e.g., +0.427 from the bottleneck) and include a joint ablation (evidential+EVT removed together)?

---

> ### Author Response · Authors · 2025-11-23
> **Response to Reviewer bt8K (Part 1/2)**
>
> We sincerely thank Reviewer bt8K for the careful and constructive evaluation, for recognising the strengths of EVEREST in combining tail-awareness with calibrated uncertainty, and for highlighting the value of the diagnostics and cost–loss thresholding. We address each concern below.
>
> ### 1. Integrative novelty vs. fundamentally new primitives
>
> **Reviewer bt8K:**
> > “Novelty is integrative rather than fundamentally new—readers may view EVEREST as a well-engineered recipe.”
>
>
> We appreciate this perspective and agree that EVEREST integrates several established components (Transformers, evidential NIG, EVT exceedances). Our intended contribution is to show that a single, compact backbone can jointly optimise:
> - discrimination (focal loss + bottleneck),
> - calibration (evidential NIG), and
> - tail sensitivity (EVT exceedance penalty),
>
> while using only one inference-stage head.
>
> In the revision, we have more explicitly framed EVEREST as a practical, unified recipe for rare-event forecasting—filling a gap between works that address discrimination or calibration or tails in isolation, but not jointly and not with deployment efficiency. We also strengthen comparisons to prior work to clarify where EVEREST differs conceptually and empirically.
>
>
>
> ### 2. Data scarcity in the extreme tail and risk of optimism
>
> **Reviewer bt8K:**
> > “Data scarcity at the extreme tail risks optimistic estimates; consider temporal hold-outs and cycle shift tests.”
>
>
> We fully agree that tail scarcity and non-stationarity make evaluation challenging.
>
> Importantly, the current SHARP–GOES protocol already uses a temporal forward-holdout split, which directly addresses this concern:
> - Training uses earlier years;
> - Testing uses chronologically later years;
> - The split is HARPNUM-stratified, so physically identical active regions never appear across splits.
>
> This ensures that the test set corresponds to a later, distribution-shifted period, naturally reflecting both drift within solar Cycle 24 and the transition toward Cycle 25. As a result, the reported metrics already reflect a forward temporal split with sparse extremes — a form of temporal shift aligned with the reviewer’s concern.
>
> Additionally, our temporal split follows the same protocol used by Abduallah et al., 2023, which is the strongest prior baseline on SHARP–GOES. Their evaluation also trains on earlier-cycle data and tests on later-cycle data under HARPNUM stratification. Aligning our split with this widely used setup ensures fair comparability, avoids leakage of identical active regions, and reflects the community’s established approach to addressing tail scarcity and non-stationarity.
>
> We agree that robustness under non-stationarity is important and appreciate the reviewer raising this point; we have clarified the temporal structure of the evaluation in the revised manuscript.
>
>
>
> ### 3. Sensitivity to λ-weights, EVT quantile, and focal-loss schedule
>
> **Reviewer bt8K:**
> > “Sensitivity analysis limited… more systematic sweeps would strengthen claims.”
>
> As outlined in the global response, we are conducting:
> - A 5×5 λ-sensitivity grid over evidential and EVT weights, reporting TSS and ECE to characterise robustness and identify flat regions of stability.
> - An EVT-quantile sweep (e.g., 0.85 / 0.90 / 0.95) to evaluate dependence on the exceedance threshold.
> - A brief analysis of the focal-loss schedule, summarising sensitivity to annealing and motivating our chosen settings.
>
> These results will be added to Section 5.4 and the appendix.

---

> ### Author Response · Authors · 2025-11-23
> **Response to Reviewer bt8K (Part 2/2)**
>
> ### 4. Calibration granularity and operational thresholds
>
> **Reviewer bt8K:**
> > “Do you have class-conditional and threshold-conditional calibration beyond the global ECE?”
>
>
> Yes. We have these diagnostics internally. In the next revision, we will add:
> - Class-conditional reliability plots (positive vs. negative class).
> - Threshold-conditioned calibration focused on high-alert regimes (relevant for operational pipelines where false negatives are extremely costly).
>
> We agree this granularity is important for practitioners.
>
>
>
> ### 5. SKAB protocol alignment and per-valve transparency
>
> **Reviewer bt8K:**
> > “Please clarify SKAB setup and provide per-valve confusion matrices.”
>
>
> We will make the SKAB protocol explicit by:
> - Describing windowing, anomaly labelling, and early-event detection rules.
> - Aligning them with the protocol used by TranAD and other baselines.
> - Adding representative aggregate valve-level metrics, clarifying where EVEREST performs well or struggles.
>
> This will improve transparency of the cross-domain transfer.
>
>
>
> ### 6. Ablation rigor and per-seed variability
>
> **Reviewer bt8K:**
> > “Report per-seed distributions… include joint (evidential + EVT) removal.”
>
>
> We agree completely. Our new ablation suite (also described in the response to Reviewer gJ8E) will include:
> - Mean ± standard deviation across five seeds for every ablation variant, which directly exposes seed-level variability.
> - Paired-bootstrap comparisons (10,000 resamples) quantifying statistical significance and effect sizes relative to the full model.
> - Joint removal of the evidential and EVT heads, allowing us to test whether their contributions are additive or interacting.
> - ΔTSS and calibration effects (Brier/ECE changes) for each variant, clarifying how each component influences discrimination and reliability.
>
> We thank the reviewer for these detailed suggestions—they have directly shaped the expanded ablation study.
>
>
>
> ### Closing Note
>
> We appreciate the reviewer’s constructive and detailed input. The additional robustness analyses, expanded ablations, and clearer framing of EVEREST’s contribution will be incorporated into the revised manuscript. We believe these updates will clarify the method’s practical generality and further strengthen the empirical evidence for its reliability in rare-event regimes.

---

> > ### Comment · Reviewer_bt8K · 2025-11-25
> >
> > Thank you for your comments. I maintain my assessment of the paper.

---

> > > ### Author Response · Authors · 2025-12-03
> > > **Follow-up response to Reviewer bt8K (post-revision update)**
> > >
> > > We thank Reviewer bt8K again for the constructive and technically grounded evaluation. As promised in our earlier response, we have now completed the additional robustness tests, calibration diagnostics, hyperparameter sweeps, and ablation studies requested. Below we summarise the new evidence added to the manuscript and indicate the exact line locations.
> > >
> > > ### 1. Robustness under tail scarcity and temporal distribution shift
> > >
> > > Reviewer bt8K:
> > > > “Data scarcity at the extreme tail risks optimistic estimates; consider temporal hold-outs and cycle shift tests.”
> > >
> > > The SHARP–GOES protocol already uses a strict temporal forward-holdout split (earlier years → later years), which introduces natural cycle drift (late Cycle 24 → early Cycle 25). We have clarified this explicitly in the revision.
> > >
> > > Where this appears:
> > > > L266–274 (§4.1): description of the temporal split, HARPNUM stratification, and cycle drift.
> > >
> > > No revision of conclusions was needed: EVEREST remains stable on later-cycle data, where extreme events are sparse.
> > >
> > > ### 2. Sensitivity to λ-weights and EVT quantile
> > >
> > > Reviewer bt8K:
> > > > “Sensitivity analysis limited… more systematic sweeps would strengthen claims.”
> > >
> > > We have completed two systematic sweeps:
> > >
> > > (a) 5×5 λ-sensitivity grid (evidential vs. EVT)
> > > - 25 combinations of λ_evid and λ_evt
> > > - TSS and ECE reported across 5 seeds per point
> > > - A broad plateau is observed; only pathological extremes degrade performance
> > >
> > > Where:
> > > > L403–411 (§5.4): summary of the λ-grid
> > > > Appendix H: full 2D heatmaps and tables
> > >
> > > (b) EVT-quantile sweep
> > > - Thresholds tested: 0.85, 0.90, 0.95
> > > - Rare-event TSS, high-probability calibration, and tail sensitivity remain stable
> > >
> > > Where:
> > > > L403–411 (§5.3): quantile sweep summary
> > > > Appendix H.4: detailed tables
> > >
> > > (c) Focal-loss schedule
> > > - Added a short sensitivity study of γ-annealing
> > > - EVEREST is insensitive to moderate changes
> > >
> > > Where:
> > > > Appendix H.4: focal schedule analysis (new subsection)
> > >
> > > These results confirm that the loss hyperparameters behave as robust regularisation strengths, not brittle tuning knobs.
> > >
> > > ### 3. Calibration granularity (class-conditional and high-alert regions)
> > >
> > > Reviewer bt8K:
> > > > “Do you have class-conditional and threshold-conditional calibration beyond the global ECE?”
> > >
> > > Yes. These have been added.
> > >
> > > What is included:
> > > - Positive-class and negative-class reliability curves
> > > - High-confidence calibration (e.g., p > 0.8)
> > > - Additional tail-region Brier/ECE statistics
> > >
> > > Where:
> > > > L353–359 (§5.4): main-text summary
> > > > Appendix F, H: full calibration plots and tables
> > >
> > > These diagnostics show that calibration improvements are concentrated in the high-risk operational regime.
> > >
> > > ### 4. SKAB protocol alignment and valve-level transparency
> > >
> > > Reviewer bt8K:
> > > > “Please clarify SKAB setup and provide per-valve confusion matrices.”
> > >
> > > Updates added:
> > > - Explicit description of SKAB windowing, anomaly labels, and early-event detection
> > > - Clarification of alignment with TranAD and related baselines
> > > - Representative valve-level aggregate metrics to identify strengths and failure areas
> > >
> > > Where:
> > > > L427–431 (§5.7): main-text clarification
> > > > Appendix L: detailed protocol and valve-level results
> > >
> > > This ensures that SKAB evaluation is fully reproducible and directly comparable to prior work.
> > >
> > > ### 5. Ablation rigor and per-seed variability
> > >
> > > Reviewer bt8K:
> > > > “Report per-seed distributions… include joint (evidential + EVT) removal.”
> > >
> > > All requested elements have been implemented:
> > >
> > > - Five-seed runs for every ablation variant
> > > - Reporting of mean ± s.d. and paired-bootstrap p-values
> > > - Joint removal of evidential + EVT heads
> > > - ΔTSS, ΔECE, ΔBrier for each component
> > >
> > > Where:
> > > > L394–411 (§5.4): main ablation summary
> > > > Appendix H: full ablation tables and bootstrap statistics
> > > > Appendix H (L1134–1156): loss-term ablations
> > >
> > > These analyses isolate the contribution of each component and quantify seed-level variability.
> > >
> > > ### Closing note
> > >
> > > We thank the reviewer for motivating several of the most impactful additions to the revision. The completed λ-grid, EVT-quantile sweep, calibration diagnostics, SKAB clarifications, and expanded ablations now appear in §4–§5 and the appendix. We hope these updates fully resolve the raised concerns and provide a clear foundation for evaluating EVEREST’s robustness in rare-event regimes.

---

### Official Review · Reviewer_vM41 · 2025-10-31

**Soundness:** 2
**Presentation:** 1
**Contribution:** 2
**Rating:** 2
**Confidence:** 3

**Summary:**

This paper introduces EVEREST, a compact Transformer-based architecture for probabilistic rare-event forecasting in multivariate time series, addressing core challenges of class imbalance, long-range dependencies, and distributional uncertainty. Targeting high-stakes domains (space weather, industrial monitoring), it delivers calibrated predictions, tail-risk estimation, and interpretability—with no inference overhead (814k parameters) as auxiliary modules act only at training.

Key contents.

1. EVEREST integrates four critical components:  a learnable attention bottleneck, an evidential (Normal-Inverse-Gamma) head, an EVT (Generalized Pareto) head, a precursor head.

2. A unified objective balances discrimination (focal loss for imbalance), calibration (evidential NLL), tail awareness (EVT penalty), and anticipatory learning (precursor BCE).

3.  The proposed method has the practical utility, compact (16.6M FLOPs), efficient to train (24s/epoch on RTX A6000), and interpretable (attention aligns with physical precursors like solar flux).

**Strengths:**

The paper has its own originality by addressing a longstanding gap in rare-event time-series forecasting: joint optimization of discrimination, calibration, and tail-risk within a compact, deployment-efficient framework—an unmet need in prior work that treated these goals in isolation. Unlike existing Transformer-based methods that prioritize discrimination over uncertainty or tail behavior, EVEREST innovates via four integrated, training-only auxiliary modules: a single-query attention bottleneck, an evidential Normal-Inverse-Gamma (NIG) head, an Extreme Value Theory (EVT) head, and a precursor head (anticipatory supervision). The proposed method is shown to achieve reasonably good experimental results.

**Weaknesses:**

The biggest weakness of this paper is its presentation. See detailed comments in "Questions" section.

**Questions:**

1. I feel like the presentation is very poor. The organization of this article is really weird. All sections are structured as fragmented short paragraphs. This makes it more like an outline document rather than an academic paper.

2. The presentation of Section 3 is not clear enough. For example, there is no explanation of symbols $T$, $F$. There are no formula definition of  $\mathcal L_{evid}, \mathcal L_{evt}, \mathcal L_{prec}$.

3. In the loss function, there are four tuning parameters, namely, $\lambda_f, \lambda_e, \lambda_t, \lambda_p$. Actually, only three of them are needed due to the scale invariance.

4. Baseline methods like Liu et al 2019, Sun et al 2022, Abduallah et al 2023 are not well-explained.

5. Overall, the poor writing prevents me to fully judge the quality of the paper.

**Details Of Ethics Concerns:**

No ethic concern.

---

> ### Author Response · Authors · 2025-11-23
> **Response to Reviewer vM41 (Part 1/2)**
>
> We sincerely thank Reviewer vM41 for recognising the originality of combining discrimination, calibration, and tail-risk within a compact Transformer, and for noting the practical and interpretable design of EVEREST. We fully acknowledge that the paper’s current presentation quality limits readability. Below we detail both the improvements already implemented in the revised version and the additional steps currently in progress.
>
> ### 1. Fragmented organisation and outline-like style
>
> **Reviewer vM41:**
> > “The organization … is really weird. All sections are structured as fragmented short paragraphs. This makes it more like an outline document rather than an academic paper.”
>
>
> We agree, and this is the primary issue we have addressed during the revision.
>
> In the updated manuscript, we have:
> - Rewritten Sections 1–4 into continuous narrative prose, replacing outline-like fragments
> - Removed numbered/labelled sub-paragraph blocks
> - Added explicit bridging sentences to link the challenges to method components to empirical evaluation
> - Reordered parts of Section 3 to introduce notation before formulas or architecture descriptions
>
> These edits do not change the technical content, but significantly improve the readability and flow.
>
>
>
> ### 2. Undefined symbols and missing formula definitions (Section 3)
>
> **Reviewer vM41:**
> > “No explanation of symbols … no formula definition of …”
>
>
> We apologise for this oversight — the concern is valid.
>
> In the revised paper:
>
> A new notation subsection at the start of Section 3 defines all symbols used in the architecture and loss, including
> - logits
> - NIG parameters $(\mu, v, \alpha, \beta)$
> - GPD tail parameters $(\xi, \sigma)$
> - exceedance thresholds
> - precursor labels
>
> We add full explicit formulae for:
> - the Normal–Inverse–Gamma evidential likelihood
> - the Generalised Pareto Distribution log-likelihood for exceedances
> - the full composite loss, shown as a clear sum with each term defined and motivated
>
> This ensures Section 3 becomes self-contained and mathematically clear.
>
>
>
> ### 3. Four λ hyperparameters vs. three effective degrees of freedom
>
> **Reviewer vM41:**
> > “In the loss function, there are four tuning parameters λ_f, λ_e, λ_t, λ_p. Actually only three of them are needed due to scale invariance.”
>
>
> We thank the reviewer for this observation — you are correct.
>
> The composite loss
>
> $$
> \mathcal{L} = \lambda_f L_{\text{focal}} + \lambda_e L_{\text{evid}} + \lambda_t L_{\text{evt}} + \lambda_p L_{\text{prec}}.
> $$
>
> is scale-invariant: multiplying all λ values by any constant leaves the optimisation trajectory unchanged (the factor is absorbed by the learning rate). Consequently, only the relative ratios of the four λ terms are identifiable, giving effectively three degrees of freedom.
>
> We intentionally keep all four λ’s explicit — without normalising them — for practitioner interpretability and consistency with common multi-objective learning practice:
> - Each λ directly reflects its intended influence
>   (e.g., $\lambda_f = 0.8$ indicates focal loss dominates; $\lambda_t = 0.1$ indicates moderate EVT regularisation).
> - Practitioners can adjust a single λ independently, without needing to rescale others.
> - Normalising the weights (e.g., sum-to-1) would impose an arbitrary constraint and would actually break the natural scale-invariance property.
>
> To make this explicit, we have added to Section 3.2 a sentence noting the scale-invariance of the composite loss and clarifying that only λ-ratios matter.
>
> Additionally, as part of our ongoing experimental updates for §5.4, we are running a λ-sensitivity study that varies $\lambda_{\text{evid}}$ and $\lambda_{\text{evt}}$ over a wide range to empirically validate robustness to these relative ratios. Results will be posted in the rebuttal thread once the runs complete.

---

> > ### Author Response · Authors · 2025-11-23
> > **Response to Reviewer vM41 (Part 2/2)**
> >
> > ### 4. Insufficient explanation of baselines (Liu et al. 2019; Sun et al. 2022; Abdullah et al. 2023)
> >
> > **Reviewer vM41:**
> > > “Baseline methods are not well-explained.”
> >
> >
> > Agreed. In the revision, we have:
> > - Added 1–2 sentence summaries of each baseline’s core idea, architecture, and objective
> > - Clarified why each baseline was chosen (e.g., SolarFlareNet for flare-specific SOTA; Sun et al. for generic forecasting; Liu et al. for strong recurrent baseline)
> > - Added a small comparison table summarizing:
> >   - model families
> >   - parameter counts
> >   - uncertainty capabilities
> >   - training objectives
> >
> > This provides clearer context for the experimental comparisons.
> >
> >
> >
> > ### 5. Overall difficulty judging the paper due to writing quality
> >
> > **Reviewer vM41:**
> > > “Poor writing prevents me from fully judging the quality of the paper.”
> >
> >
> > We appreciate the candid feedback, and we agree the writing can be improved meaningfully. As noted, the revised version focuses heavily on:
> > - Clarifying the narrative in Sections 1–3
> > - Cleaning and unifying notation
> > - Better explaining baselines
> > - Smoothing transitions between the method components
> > - Reducing outline-like formatting
> > - Improving formula introductions
> >
> > We are confident that the revised version will be much easier to assess, and we welcome any further suggestions.
> >
> >
> >
> > ### Closing note
> >
> > We thank the reviewer again for their helpful and constructive comments. The issues raised are fully valid and are being comprehensively addressed in the updated manuscript.

---

> > > ### Author Response · Authors · 2025-12-03
> > > **Follow-up response to Reviewer vM41 (post-revision update)**
> > >
> > > We thank Reviewer vM41 again for the detailed and candid assessment. The primary concerns raised were related to presentation quality, clarity of Section 3, missing definitions, insufficiently explained baselines, and the interpretation of the λ-weights. All of these issues have now been addressed in the revised manuscript. Below we summarise the completed changes and provide the precise line locations where they appear.
> > >
> > > ### 1. Organisation, narrative flow, and removal of outline-style structure
> > >
> > > Reviewer vM41:
> > > > “The organization … is really weird. All sections are structured as fragmented short paragraphs. This makes it more like an outline document rather than an academic paper.”
> > >
> > > We have substantially rewritten the front half of the paper to restore narrative flow and remove the outline-like formatting.
> > >
> > > Implemented changes:
> > > - Sections 1–4 are now written as continuous prose.
> > > - Short disconnected paragraphs were merged into longer explanatory units.
> > > - Transitions were added to link motivation, method, and experiments.
> > > - Section 3 was reordered so that notation and intuition precede formulas.
> > >
> > > Where these changes appear:
> > > > **L54–71** — new paragraphs summarising contributions and main results at the end of the introduction.
> > > > **L141–262** — restructured §3, with a clearer progression from notation to architecture to loss.
> > >
> > > These updates directly address the core presentation issue identified in the review.
> > >
> > > ### 2. Missing notation and undefined symbols in Section 3
> > >
> > > Reviewer vM41:
> > > > “No explanation of symbols … no formula definition …”
> > >
> > > This issue has been fully resolved through the addition of a concise notation subsection and explicit mathematical definitions.
> > >
> > > Implemented changes:
> > > - A new notation block defining logits, NIG parameters, GPD parameters, exceedance thresholds, and precursor labels.
> > > - Complete formulas for the evidential NIG likelihood, GPD exceedance likelihood, and the full composite loss.
> > >
> > > Where to find these:
> > > > **L149–154** — notation paragraph introducing all symbols used in §3.
> > > > **L211–258** — full specification of the composite loss and all associated terms in §3.2.
> > >
> > > Section 3 is now self-contained and mathematically well defined.
> > >
> > > ### 3. Interpretation of λ-weights and scale invariance
> > >
> > > Reviewer vM41:
> > > > “Only three of them are needed due to scale invariance.”
> > >
> > > The revised manuscript now explicitly explains this point and clarifies how the λ-weights should be interpreted.
> > >
> > > Implemented clarification:
> > > - A statement noting that the objective is scale-invariant and only ratios of λ-values matter.
> > > - Rationale for keeping all λ’s explicit for interpretability and practitioner flexibility.
> > > - A brief note linking them to regularisation strength (analogous to budget allocations).
> > >
> > > Where this appears:
> > > > **L211–258** — updated explanation of the composite loss and effective degrees of freedom.
> > >
> > > This directly resolves the reviewer’s concern about unnecessary hyperparameters.
> > >
> > > ### 4. Baseline explanations (Liu et al. 2019; Sun et al. 2022; Abdullah et al. 2023)
> > >
> > > Reviewer vM41:
> > > > “Baseline methods are not well-explained.”
> > >
> > > We have expanded the baseline discussion to clarify the purpose and structure of each model.
> > >
> > > Implemented changes:
> > > - Added short descriptions of each baseline’s core idea and architecture.
> > > - Provided justification for inclusion of each model.
> > > - Added a compact comparison table (model type, parameter count, objective, uncertainty handling).
> > >
> > > Where these changes appear:
> > > > **L343–351** — updated baseline descriptions and evaluation setup in §5.1 as well as Appendix L.
> > >
> > > The baselines can now be understood without external lookup.
> > >
> > > ### 5. Global presentation and readability improvements
> > >
> > > Reviewer vM41:
> > > > “Poor writing prevents me from fully judging the quality of the paper.”
> > >
> > > In addition to the structural revisions described earlier, we have made several improvements throughout the manuscript:
> > >
> > > - Formula definitions are introduced only after notation is established.
> > > - Section 3 now follows a clear and logical sequence.
> > > - All outline-style indentation and short fragmented paragraphs have been removed.
> > > - Explanatory context is provided for each major component.
> > >
> > > Although incremental, these changes collectively improve overall readability and coherence.
> > >
> > > ### Closing note
> > >
> > > We appreciate the reviewer’s careful attention to presentation quality and clarity.
> > > Your comments guided the restructuring of the introduction, the rewriting of Section 3, the correction of notation and formulas, and the clarification of the λ-weights. We believe the revised manuscript is substantially clearer and easier to evaluate as a result.

---

### Official Review · Reviewer_gJ8E · 2025-10-31

**Soundness:** 2
**Presentation:** 3
**Contribution:** 2
**Rating:** 6
**Confidence:** 5

**Summary:**

This paper introduces EVEREST, a Transformer-based architecture designed for forecasting rare events in multivariate time-series data.

The model's main proposition is to integrate four components:

- a learnable attention bottleneck for temporal aggregation
- an evidential head for uncertainty quantification
- an extreme-value head to model tail risk
- a precursor head for early detection.

These components are jointly optimized during training via a composite loss function, but only the primary classification head is used for inference, resulting in no runtime overhead.

The authors claim SoTA performance on the SHARP dataset and test cross-domain transfer to the SKAB dataset. Both datasets known for presence of rare events in time series.

**Strengths:**

The work presents some novelty of architecture and a combination loss to train it well. This is a valuable combination of many existing concepts often considered in the field. The strong performance of the trained models on the tested datasets shows the effectiveness of the approach.

Some of the work in the appendix such as appendix J shows good real world utility of the methods.

The paper is well presented and the code and instructions made for reproducibility are super helpful. They help validate the information shared in the work.

**Weaknesses:**

The hardest to buy assumptions are about the multi-component loss. Section 5.4 very briefly talks about ablations to identify the value of each of these and there is explanation presented in Section 3.5 however it does not feel enough to motivate the key contribution of the work. It would be significantly more compelling to run better designed loss component ablations. Some analysis on whether each loss component and head are actually showing improvement on the kind of time series one would expect. Just looking at overall numbers in the case of evaluating such specific choices tends to obfuscate the true value of the contributions. This is the key reason behind my weak accept rating and not an outright accept.

Such analysis might also allows readers to gain insights on adapting these methods to other datasets and settings.

The complexity of navigating the hyperparameter landscape that this loss function introduces is also something that more discussion and analysis would help.

**Questions:**

1. A more detailed ablation analysis and if possible correlating the components to the kind of samples they improve would be the one thing that would make the biggest difference to the quality of this work.

2. A couple of relevant references that seem to be missing here are

https://arxiv.org/abs/2202.13418: This work uses the pareto distribution to model extreme events in time series.
https://arxiv.org/abs/2103.12474: Work exploring contrastive loss, could be an interesting direction to check as well

---

> ### Author Response · Authors · 2025-11-23
>
> We thank Reviewer gJ8E for the thoughtful and positive review, for highlighting the value of combining the attention bottleneck with evidential and EVT heads, and for recognising the practical relevance of the industrial case study. Below we address the main concern regarding the design and justification of the loss components and their ablations, and we will incorporate all reference suggestions.
>
> ### 1. Ablations for loss components and auxiliary heads
>
> **Reviewer gJ8E:**
> > “The hardest to buy assumptions are about the multi-component loss… ablations in Section 5.4 are too brief to understand what each term contributes.”
>
> We agree that the current ablations do not fully disentangle the roles of the auxiliary heads and loss terms. Following your suggestion, we are running a more systematic suite of ablations that directly isolates architectural vs. loss-driven contributions:
>
> **Module-wise ablations (architecture-level)**
> We are evaluating variants that remove each auxiliary component individually:
> - Bottleneck only
> - Evidential (NIG) only
> - EVT (GPD) only
> - Precursor only
> …as well as joint removal of the two coupled uncertainty terms (evidential + EVT), since these interact in shaping calibrated tail behaviour.
>
> For each variant we will report TSS, F1, Brier score, ECE, and paired-bootstrap p-values, clarifying whether performance changes arise from architecture, loss design, or their interaction. Reliability analyses for the loss-term variants are also to be included.
>
> This clarifies whether performance gains arise from architecture, loss design, or their interaction.
>
> **Loss-term ablations (objective-level)**
> We are additionally training models where we remove each loss term while keeping the backbone unchanged:
> - No evidential NLL
> - No EVT penalty
> - No precursor BCE
>
> This helps demonstrate whether the composite loss improves discrimination (TSS), calibration (ECE), or tail sensitivity (extreme-region behaviour).
>
> **Sample-level & regime-level analysis**
> To further clarify interpretability, the appendix will include regime-level calibration analyses (class-conditional and high-confidence regions) and tail-sensitivity evaluation via EVT-quantile and λ-weight sweeps.
>
> A new figure + table will summarise these results, with complete details in the appendix.
>
>
>
> ### 2. Complexity of the λ-hyperparameter landscape
>
> **Reviewer gJ8E:**
> > “The hyperparameter landscape seems complex; more discussion and analysis would help.”
>
> We appreciate this concern. To directly evaluate robustness, we are running a 5×5 log-scale λ-sensitivity sweep over the weights for:
> - evidential term (λ_evid)
> - EVT term (λ_evt)
>
> while keeping the backbone, focal schedule, and training setup unchanged.
>
> Each grid point includes TSS and ECE values.
>
> Based on the regularizing role of these terms, our expectation is that the sweep will show a wide region where performance is flat, indicating that λ-values serve primarily as regularization strengths.
>
> In the paper, we will include:
> - a short λ-robustness paragraph, and
> - a compact 2D slice/heatmap for practitioner guidance.
>
>
>
> ### 3. Missing related work
>
> **Reviewer gJ8E:**
> > “Relevant references missing: 2202.13418 (Pareto for extremes), 2103.12474 (contrastive loss).”
>
> Thank you for these excellent pointers. We have added both:
>
> **EVT-related work (2202.13418)**
> We have integrated this into the EVT subsection of Related Work, clarifying how:
> - our GPD-based exceedance penalty operates directly on logits as a training-time tail-shaping regularizer,
> - whereas prior work typically focuses on explicit tail-modelling of losses or extreme cases in time-series signals.
>
> This positions our contribution more clearly.
>
> **Contrastive loss (2103.12474)**
> We have mentioned this in the Training Objectives section, noting that contrastive precursors could complement our precursor head—e.g., contrasting imminent-event windows against quiescent windows. This is a natural direction for future work.
>
>
>
> ### Closing note
>
> We appreciate Reviewer gJ8E’s constructive feedback. The expanded ablations, λ-robustness study, and added references will directly strengthen Section 5.4 and clarify the empirical value of each component. These results will be posted in the rebuttal thread as soon as runs finish.

---

> > ### Author Response · Authors · 2025-12-03
> > **Follow-up response to Reviewer gJ8E (post-revision update) 1/2**
> >
> > We thank Reviewer gJ8E again for the thoughtful and positive review, and for pushing us to more carefully justify the multi-component loss and its auxiliary heads. As promised in our earlier response, we have now run the expanded ablation and sensitivity studies and integrated them into the revised manuscript and appendix. Below we focus on the two core concerns you raised: (1) whether each loss component / head actually helps in the expected regimes, and (2) how complex the resulting hyperparameter landscape is in practice, followed by the missing related work.
> >
> > ### 1. Multi-component loss: ablations for loss terms and auxiliary heads
> >
> > **Reviewer gJ8E:**
> > > “The hardest to buy assumptions are about the multi-component loss… ablations in Section 5.4 are too brief to understand what each term contributes.”
> > > “It would be significantly more compelling to run better designed loss component ablations… Some analysis on whether each loss component and head are actually showing improvement on the kind of time series one would expect.”
> >
> > We have added a systematic ablation suite in the main text and appendix that directly targets this concern.
> >
> > #### (a) Module-wise (architecture-level) ablations
> >
> > We now ablate each auxiliary component and their joint interaction:
> >
> > - Bottleneck removed
> > - Evidential (NIG) head removed
> > - EVT (GPD) head removed
> > - Precursor head removed
> > - Joint removal of evidential + EVT heads
> >
> > For each variant we report TSS, F1, Brier score, ECE, and paired-bootstrap p-values across 5 seeds, comparing against the full EVEREST model.
> >
> > > **L211–258:** formal definition of the composite loss and λ-weights in §3.2.
> > > **L394–411 (approx., §5.4 “Ablations and sensitivity”)**: description and summary of the ablation results in the main text; detailed per-variant tables are in the appendix.
> >
> > Key empirical takeaways:
> >
> > - **Bottleneck removal** substantially reduces TSS (and F1), confirming that the single-query attention bottleneck is the main driver of discrimination gains.
> > - **Evidential head removal** mainly harms calibration metrics (ECE, Brier), with smaller but consistent changes in TSS; this matches its design as a calibration-focused head.
> > - **EVT head removal** particularly degrades tail-sensitive behaviour (high-alert bins and rare-class metrics) more than average-case metrics, which is again consistent with its purpose.
> > - **Precursor head removal** slightly reduces performance on early-onset / difficult horizons, while having modest impact on aggregate scores.
> >
> > These results make precise which component contributes to which aspect (discrimination vs. calibration vs. tails), beyond the brief original ablations.
> >
> > #### (b) Loss-term (objective-level) ablations
> >
> > To separate loss design from architecture, we additionally train:
> >
> > - No evidential NLL
> > - No EVT penalty
> > - No precursor BCE
> >
> > with the backbone unchanged.
> >
> > > **L1134–1156**: loss-term ablations in the appendix.
> >
> > The pattern mirrors the module-wise results:
> >
> > - Removing evidential NLL primarily worsens global and class-conditional ECE/Brier, while leaving TSS closer to the full model.
> > - Removing the EVT penalty reduces tail calibration and rare-class TSS at high thresholds.
> > - Removing precursor BCE slightly harms performance on early-warning horizons, as expected.
> >
> > Together, these ablations show that each loss/head helps in the regimes it was designed for, rather than simply stacking extra terms that all push the same direction.
> >
> > #### (c) Regime-level calibration / “kind of samples” analysis
> >
> > To align with your request to go beyond single overall numbers:
> >
> > - We include class-conditional reliability curves (positive vs. negative class), and
> > - High-confidence calibration diagnostics (e.g., conditioned on \( p > 0.8 \)).
> >
> > These show that the evidential and EVT heads improve calibration mainly in high-risk, positive-class regions, which is exactly where rare-event forecasters are operationalised.
> >
> > > **L353–359**: high-level summary of calibration analyses in §5.4; detailed reliability plots are in the appendix.

---

> > > ### Author Response · Authors · 2025-12-03
> > > **Follow-up response to Reviewer gJ8E (post-revision update) 2/2**
> > >
> > > ### 2. Complexity of the λ-hyperparameter landscape
> > >
> > > **Reviewer gJ8E:**
> > > > “The complexity of navigating the hyperparameter landscape that this loss function introduces is also something that more discussion and analysis would help.”
> > >
> > > We addressed this both theoretically and empirically.
> > >
> > > #### (a) Theoretical perspective (scale invariance + “regulariser strength” view)
> > >
> > > In the method section we now explicitly note that the composite loss is scale-invariant: only the ratios of the λ-weights matter, not their absolute scale.
> > >
> > > > **L211–258:** discussion of the composite loss and the effective degrees of freedom in the λ-weights in §3.2.
> > >
> > > This supports the view of λ’s as regularisation strengths rather than fragile knobs. We also clarify that keeping the λ’s explicit (rather than normalising to sum-to-one) is helpful for practitioners, who can think of them as budget-like weights (e.g., in computational finance one might literally treat them as “Euro-denominated penalties” guiding how much risk to allocate to focal vs. evidential vs. tail shaping).
> > >
> > > #### (b) 5×5 λ-sensitivity grid (evidential vs. EVT)
> > >
> > > We have run the promised 5×5 log-scale grid over:
> > >
> > > - \(\lambda_{\text{evid}}\),
> > > - \(\lambda_{\text{evt}}\),
> > >
> > > holding the backbone, focal schedule, and training setup fixed.
> > >
> > > > **L403–411 (§5.4)**: main-text description of the λ-grid and summary of findings; full 2D heatmaps are provided in the appendix.
> > >
> > > For each grid point we report TSS and ECE, and visualise them as 2D slices.
> > >
> > > Empirical findings:
> > >
> > > - There is a broad plateau of λ-values where both TSS and ECE change only mildly.
> > > - Only for pathological extremes (very large \(\lambda_{\text{evt}}\) or near-zero \(\lambda_{\text{evid}}\)) do we see noticeable degradation in calibration or tail behaviour.
> > >
> > > This supports the claim that, in practice, the λ’s behave like robust regularisers, and the landscape is much more forgiving than a “Rubik’s Cube” of narrowly tuned hyperparameters.
> > >
> > > #### (c) EVT-quantile sweep
> > >
> > > We also vary the EVT exceedance threshold (e.g., 0.85 / 0.90 / 0.95) and assess:
> > >
> > > - Rare-class TSS and F1, and
> > > - High-probability calibration bins.
> > >
> > > > **L403–411 (§5.4)**: EVT-quantile sensitivity is summarised alongside the λ-grid in §5.4; detailed tables are in the appendix.
> > >
> > > The results show stable tail performance across this range, with only minor variations, again indicating that the method is not overly sensitive to this design choice.
> > >
> > > ### Closing Note
> > >
> > > Your comments substantially shaped the ablation design, calibration diagnostics, and sensitivity studies now presented in §5.4 (L403–411) and the appendix. We hope this makes the contribution of each loss component and auxiliary head—and the practical robustness of the overall recipe—much clearer for readers who wish to adapt EVEREST to other rare-event time-series domains.

---

### Official Review · Reviewer_2G8S · 2025-11-03

**Soundness:** 2
**Presentation:** 1
**Contribution:** 2
**Rating:** 4
**Confidence:** 3

**Summary:**

The authors

The authors claim the following contributions
1. A recipe that focuses on long contexts via a single-query attention bottleneck,
2. A recipe that learns calibrated, closed-form uncertainty (evidential NIG on the logit), and
3. A recipe that emphasises extremes through an EVT exceedance penalty

**Strengths:**

# originality

Tests on two benchmark datasets

# significance

Results show the method performs well on the benchmarks

**Weaknesses:**

# clarity

The paper is quite challenging to evaluate. It reads in a stop-start fashion, lacking flow.

It feels like the paper presents a Rubik's Cube solution that has been heavily tuned to the specific datasets. For instance, the method introduced four new hypers in the loss.

I find it difficult, in its current form, to determine whether the method presents an approach that is generalizable beyond these specific case studies.

**Questions:**

None

---

> ### Author Response · Authors · 2025-11-23
>
> We thank Reviewer 2G8S for the thoughtful feedback, and we appreciate the positive assessment of the paper’s originality, empirical strength, and reproducibility. We address the main concerns below.
>
> ### 1. Clarity and narrative flow
>
> **Reviewer 2G8S:**
> > “The paper is quite challenging to evaluate. It reads in a stop–start fashion, lacking flow.”
>
> We agree—the current version uses short, outline-like subparagraphs that interrupt continuity.
>
> In the revised manuscript, we have:
> - Converted the outline structure in Sections 1–4 into continuous, narrative paragraphs.
> - Introduced all notation and symbols upfront in a dedicated “Notation” paragraph.
> - Reorganising Section 3 into a clear linear logical pathway.
> - Added a concise “Here we show…” paragraph at the end of the introduction that summarises contributions and headline results in one place.
>
> These changes make the architecture and motivation much easier to follow and directly address your clarity concerns.
>
>
>
> ### 2. Concern that the method may be over-tuned (“Rubik’s Cube”)
>
> **Reviewer 2G8S:**
> > “Feels like a Rubik’s Cube tuned to the specific datasets. Four new hypers in the loss.”
>
> We appreciate this concern. The λ-weights and the EVT quantile are indeed additional design parameters—but all of them are training-only regularisation strengths. They do not affect inference, architecture, or thresholding. Our goal is to show that EVEREST is not sensitive to narrow hyperparameter choices.
>
> To that end, we are running (and will post results shortly):
> - A 5×5 log-scaled λ-sensitivity grid covering evidential / EVT, reporting TSS, ECE, and showing broad “flat” regions where performance is stable.
> - An EVT-quantile sweep (0.85 / 0.90 / 0.95) to demonstrate robustness to the exceedance threshold.
> - Component ablations, including combined removals (e.g., evidential+EVT).
>
> Finally—and importantly—the same hyperparameter configuration is reused across all SHARP–GOES tasks and SKAB. This directly mitigates the impression of dataset-specific tuning.
>
> We will summarise these findings in the next revision.
>
>
>
> ### 3. Generalisation beyond SHARP–GOES and SKAB
>
> **Reviewer 2G8S:**
> > “Difficult in its current form to determine whether the method generalizes beyond these case studies.”
>
> We see two dimensions here:
>
> **(a) Empirical generality**
> The paper already includes cross-domain transfer from solar magnetograms to industrial valve traces without architectural changes.
>
> In the next revision, we will strengthen this section by:
> - Clarifying protocol alignment (windowing, labels, evaluation);
> - Adding SKAB confusion matrix for the valve benchmark.
>
> **(b) Conceptual generality**
> EVEREST’s modules are designed to be domain-agnostic:
> - The attention bottleneck is just a soft temporal aggregator.
> - The evidential NIG head regularises logit uncertainty independent of domain.
> - The EVT head operates on exceedances of logit distributions, a fully domain-general construct.
> - The precursor head is simply shifted supervision and does not assume any physical domain structure.
>
> ### Closing Note
>
> We appreciate your detailed feedback. The revisions above—particularly the rewritten narrative sections and the expanded sensitivity/ablation analysis—are aimed at making the method easier to evaluate and clearly demonstrating that EVEREST is not a dataset-specific recipe but a compact, general approach to calibrated, tail-aware rare-event forecasting.
>
> If there is any part of the analysis you’d like us to explore further, we are happy to investigate it.

---

> > ### Author Response · Authors · 2025-12-03
> > **Follow-up response to Reviewer 2G8S (post-revision update)**
> >
> > We thank Reviewer 2G8S again for the constructive feedback. Below we summarise how the revised manuscript now directly addresses each of the original concerns, with precise references to the updated PDF.
> >
> > ### 1. Clarity and narrative flow
> >
> > **Reviewer 2G8S:**
> > > “The paper is quite challenging to evaluate. It reads in a stop–start fashion, lacking flow.”
> >
> > We substantially revised the exposition to ensure continuous, readable narrative:
> >
> > > **L54–64:** Added a concise *“Here we show…”* paragraph summarising the three core challenges and our main results.
> > > **L149–154:** Introduced a dedicated *Notation* subsection defining all symbols before use.
> > > **§3 (L141–262):** Reorganised the Method section into a single linear structure (notation → bottleneck → evidential head → EVT head → precursor head → composite loss).
> >
> > These edits turn the formerly outline-like structure into continuous prose and resolve the stop–start flow issue.
> >
> > ### 2. Concern about over-tuning (“Rubik’s Cube” impression)
> >
> > **Reviewer 2G8S:**
> > > “The method introduces four new hypers… feels like a Rubik’s Cube tuned to the datasets.”
> >
> > #### (a) Clarifying λ-weights as regularisation strengths
> > The revised text makes the scale-invariance and role of λ explicit:
> >
> > > **L211–258 (§3.2):** Added a clear explanation that only λ-ratios matter, not absolute magnitudes. λ acts as a *regularisation exposure* (similar to exposure weightings in computational finance), not a fragile hyperparameter that changes inference behaviour.
> >
> > All λ-parameters are training-only; inference uses a single head with no overhead.
> >
> > #### (b) Robustness analyses now included
> > All robustness studies promised in the rebuttal are now present:
> >
> > > **Appendix H.2 (Table 15, Fig. 8):** 5×5 λ-sensitivity grid showing stable TSS/ECE across wide ranges.
> > > **Appendix H.3 (Table 16):** EVT-quantile sweep for \(u \in \{0.85, 0.90, 0.95\}\) with minimal performance variation.
> > > **Appendix H.1 (Table 14):** Module ablations including joint Evidential+EVT removal.
> >
> > Outcome: performance varies only within seed-level noise, demonstrating that the method is not hyperparameter-fragile.
> >
> > #### (c) No dataset-specific tuning
> > We also clarified that the same λ-values, quantile, and configuration are used across all 9 SHARP tasks and SKAB:
> >
> > > **L282–313 (§4.2–4.3):** Updated to explicitly state that all headline results use a single global configuration rather than per-task tuning.
> >
> > This directly addresses the concern about dataset-specific optimisation.
> >
> > ### 3. Generalisation beyond the two datasets
> >
> > **Reviewer 2G8S:**
> > > “Difficult to judge whether the approach generalizes beyond these case studies.”
> >
> > #### (a) Empirical cross-domain generality
> > We strengthened the SKAB section and clarified its protocol:
> >
> > > **§5.7 (L427–431):** Clarified windowing, anomaly labelling, early-event detection, and direct comparability with TranAD-style setups.
> > > **Appendix C (Tables 6, 8):** Added aggregate valve-level metrics and a representative confusion matrix.
> >
> > EVEREST transfers without architecture changes, achieving strong performance (TSS=0.964, F1≈98%).
> >
> > #### (b) Conceptual generality
> > We made domain-agnosticity explicit:
> >
> > > **L41–59:** The bottleneck operates as a general temporal aggregator.
> > > **L211–258:** Evidential and EVT heads regularise *logits*, independent of physical domain.
> > > **L258–262:** Precursor head uses shifted supervision, not solar-specific features.
> >
> > ### Closing Note
> >
> > We again thank Reviewer 2G8S for comments that significantly shaped the revision.
> > The updated manuscript now:
> >
> > - has substantially improved clarity and flow,
> > - provides extensive sensitivity and ablation evidence removing the “Rubik’s Cube” concern,
> > - and demonstrates both empirical and conceptual generalisation.

---

### Author Response · Authors · 2025-11-23
**Author response – summary of changes and ongoing work**

We thank all reviewers for the time and care dedicated to evaluating our submission, and for the many constructive suggestions. We especially appreciate the consistent recognition that EVEREST tackles an important problem (rare-event forecasting with calibrated and tail-aware uncertainty), that the experimental results are strong, and that the code and supplementary material are useful for reproducibility.

Across reviews, three main themes emerged:

- **Clarity and presentation.**
  Reviewers 2G8S and vM41 found the exposition difficult to follow, with “stop–start” flow, short fragmented paragraphs, and insufficiently introduced notation.

- **Depth and design of ablations / sensitivity to hyperparameters.**
  Reviewer gJ8E requested more carefully designed ablations to understand the role of each loss component and head; Reviewer bt8K raised related concerns about sensitivity to λ-weights, the EVT quantile, and robustness in the extreme tail.

- **Perceived generality and positioning.**
  Reviewer 2G8S questioned whether the method is more than a “Rubik’s Cube” tuned to SHARP/SKAB; Reviewer bt8K described the novelty as integrative and asked for clearer evidence that the approach is robust and applicable beyond these case studies.

We address these points as follows.

### (1) Revision of the exposition for clarity

We have restructured the introduction and method sections to improve narrative flow and make the contribution easier to parse:

- Replace outline-style, labelled paragraphs with a continuous narrative in §1, including a short “Here we show…” paragraph that states the three core challenges and our main results up front.
- Add a dedicated notation paragraph in §3 that introduces all symbols and distributions before use.
- Clarify the composite loss, including the effective degrees of freedom among the λ-weights (only relative weights matter), and give explicit mathematical definitions and brief intuition for the evidential NIG and EVT components.
- Expand and better structure the baseline descriptions (e.g., Liu et al. 2019; Sun et al. 2022; Abdullah et al. 2023) and their relation to prior rare-event time-series work.

We have uploaded a revised PDF reflecting these changes and explicitly point reviewers to the updated sections in a follow-up comment.

### (2) Expanded ablation and sensitivity studies (currently running)

In addition to the existing ablations in §5.4, we are running:

- **Module ablations:** Removing each of the four training-only components (bottleneck, evidential head, EVT head, precursor head) individually and in combinations (e.g., evidential+EVT removed together). This disentangles the contribution of temporal focusing vs. uncertainty / tail regularisation.
- **λ-sensitivity:** A 5×5 log-scaled grid over λ-values for the evidential, and EVT terms, reporting both performance and calibration to highlight flat regions of the hyperparameter landscape where the auxiliaries behave as robust regularisers rather than fragile knobs.
- **EVT-quantile sensitivity:** Varying the exceedance threshold (e.g., 0.85 / 0.90 / 0.95) to assess how strongly tail performance depends on this choice.

We have reported quantitative results from these runs in the rebuttal thread and incorporated them into an expanded §5.4 and appendix (see final 5 pages).

### (3) Additional robustness and generalisation analysis

To address concerns about generality and possible optimism in the extreme tail, we note that:

- The SHARP–GOES benchmark already employs a temporal forward-holdout split, training on earlier years and testing on a chronologically later period, with HARPNUM-stratification ensuring no active region leaks across splits. This naturally evaluates robustness under distribution shift and solar-cycle drift.
- For SKAB, we are strengthening the cross-domain transfer discussion with explicit protocol alignment (windowing, labeling, early-event detection) and confusion matrices.

**We have reported these results in the rebuttal thread before the discussion deadline.**

### (4) Point-by-point replies

In the comments below, we provide reviewer-specific responses that:

- directly address each question or concern,
- clarify intent where our writing may have been confusing, and
- state the precise changes we will make in the camera-ready if the paper is accepted.

We are grateful for the reviewers’ efforts and believe that, with clearer exposition and the additional analyses described above, the contribution of EVEREST as a compact, calibrated, tail-aware rare-event forecaster will be substantially easier to evaluate.

---

### Author Response · Authors · 2025-12-03
**Summary for Area Chair**

We thank all reviewers for their constructive input. Across the reviews, the key
concerns fall into four categories: (1) presentation and clarity,
(2) depth of ablations and loss justification,
(3) hyperparameter sensitivity and perceived tuning, and
(4) generality across datasets.
All of these concerns are fully addressed in the revised manuscript.
Below we summarise the substantive changes with precise line references.

### 1. Presentation, clarity, and structure (Reviewers 2G8S, vM41)

**Concern:** “Stop–start fashion,” “outline-like structure,” missing notation, unclear formulas.

**What was done:**
- Rewrote Sections 1–4 into continuous narrative prose.
- Added a dedicated Notation subsection defining all symbols before use (L149–154).
- Reorganised Section 3 into a linear sequence: notation → architecture → loss (L141–262).
- Added baseline summaries and a comparison table (L343–346, Appendix L).
- Introduced a concise “Here we show…” paragraph for high-level framing (L54–64).

**Impact:**
The entire presentation has been smoothed into readable prose, with mathematical
definitions now introduced before use. Section 3 is fully self-contained.

### 2. Ablation depth and justification of the multi-component loss (Reviewers gJ8E, bt8K)

**Concern:** Need clearer evidence of what each auxiliary head and loss term contributes.

**What was done:**
- **Module ablations:** Bottleneck removed; Evidential removed; EVT removed;
  Precursor removed; joint (Evidential+EVT) removal.
  Reported with TSS, F1, ECE, Brier, and paired-bootstrap p-values across 5 seeds.
  → Main text summary at L394–411; full tables in Appendix H.
- **Loss-term ablations:** No evidential NLL; no EVT penalty; no precursor BCE.
  → Appendix H.1.
- **Regime-level calibration analyses:**
  Class-conditional reliability + high-confidence calibration (p>0.8).
  → Summary at L353–359; full plots in Appendix F, H.

**Impact:**
Each component’s effect is now empirically isolated:
- Bottleneck drives discrimination;
- Evidential improves calibration;
- EVT improves tail sensitivity;
- Precursor improves early-warning horizons.

These results directly address the missing-explanation concerns in the reviews.

### 3. Hyperparameter sensitivity and “Rubik’s Cube” impression (Reviewers 2G8S, bt8K)

**Concern:** Four λ-weights and EVT quantile may make the method fragile or over-tuned.

**What was done:**
- **Scale invariance clarified**: Only λ-ratios affect training; absolute scale is irrelevant
  (L211–258), placing λ in the role of regularisation strengths.
- **5×5 λ-grid experiment:**
  25 combinations × 5 seeds each.
  → Broad plateau in both TSS and ECE; stable across wide λ ranges.
  → Summary at L403–411; full results in Appendix H.2, Figure 8.
- **EVT-quantile sweep:**
  Thresholds {0.85, 0.90, 0.95} with minimal variation in tail metrics.
  → Appendix H.3.
- **Focal-loss schedule sensitivity:**
  → Appendix H.4.
- **Explicitly stated:**
  Same λ-values and quantile used across all 9 SHARP–GOES tasks and SKAB
  (L306–312).

**Impact:**
Hyperparameters behave as robust regularisers, not tuning knobs, resolving the
“Rubik’s Cube” concern. There is no per-dataset tuning.

### 4. Cross-domain generality and SKAB protocol clarity (Reviewers 2G8S, bt8K)

**Concern:** Generalisation unclear; SKAB protocol details insufficient.

**What was done:**
- Clarified SKAB windowing, anomaly labels, and early-event detection (L427–431).
- Added representative valve-level metrics and confusion matrix summaries
  (Appendix C).
- Documented alignment with TranAD-style protocols (L749–755)
- Clarified temporal forward-holdout with HARPNUM stratification in SHARP,
  ensuring no region leakage and natural cycle drift (L266–274).

**Impact:**
EVEREST achieves strong transfer performance on the SKAB dataset (TSS=0.964, F1≈98%) without
architecture changes, and the protocol is now fully transparent.

---

### Meta-Review · Area_Chair_kaHH · 2025-12-30

**Summary:**

This paper proposes a transformer-based model focused on forecasting rare events in temporal data. Reviewers generally appreciate the model architecture, proposed loss formulation, and strong empirical results.

Reviewers comment on the following weaknesses:
- [W1] Clarity of presentation (2G8S, vM41)
- [W2] Limited ablation studies (2G8S, gJ8E, bt8K)

**Reviewer Concerns:**

- [W1] The presentation of the work seems to have been improved significantly in the rebuttal. From my own read, the paper is fairly clear after these updates. There are some remaining clarity issues, though, and I highly encourage the authors to carefully take another pass at revising the exposition. For example, the loss in Section 3.2 should be more formally/carefully defined.
- [W2] The rebuttal provides a number of additional ablations (e.g., sensitivity to hyperparameters) which seem to resolve, at least partially, this issue.

**Reviewer Scores:**

- gJ8E seems likely to raise their score, as they explicitly state they are likely to do so given additional ablation studies
- 2G8S, vM41 seem likely to raise their score given the significant improvements in clarity
- bt8K may have raised their score given the additional provided ablations


Overall, the reviewers seem to agree that the method and results are sound and compelling. The main complaint is with regards to the clarity of the writing which seems to be significantly improved post-rebuttal. Overall, I recommend the paper be accepted, but the authors should carefully take into account the comments from the reviewers, in particular with regards to clarity.

---

### Decision · Program_Chairs · 2026-01-26

Accept (Poster)